# 100 years of anthropogenic impact causes changes in freshwater functional biodiversity

**Niamh Eastwood[1]\*[†], Jiarui Zhou[1][†], Romain Derelle[1], Mohamed Abou-Elwafa Abdallah[2], William A Stubbings[1,2], Yunlu Jia[3], Sarah E Crawford[3], Thomas A Davidson[4], John K Colbourne[1], Simon Creer[5], Holly Bik[6], Henner Hollert[3,7,8], Luisa Orsini[1,9]**

[1]Environmental Genomics Group, School of Biosciences, University of Birmingham, Birmingham, United Kingdom; [2]School of Geography, Earth & Environmental Sciences, University of Birmingham, Birmingham, United Kingdom; [3]Department Evolutionary Ecology & Environmental Toxicology, Faculty of Biological Sciences, Goethe University Frankfurt, Frankfurt, Germany; [4]Lake Group, Department of Ecoscience, Aarhus University, Aarhus, Denmark; [5]School of Natural Sciences, Environment Centre Wales, Deiniol Road, Bangor University, Bangor, United Kingdom; [6]Department Marine Sciences and Institute of Bioinformatics, University of Georgia, Athens, United States; [7]LOEWE Centre for Translational Biodiversity Genomics (LOEWE-TBG), Frankfurt, Germany; [8]Department Media-related Toxicology, Institute for Molecular Biology and Applied Ecology (IME), Frankfurt, Germany; [9]The Alan Turing Institute, British Library, London, United Kingdom

**\*For correspondence:**
niamh.m.eastwood@gmail.com

[†]These authors contributed equally to this work

**Competing interest:** The authors declare that no competing interests exist.

**Abstract** Despite efforts from scientists and regulators, biodiversity is declining at an alarming rate. Unless we find transformative solutions to preserve biodiversity, future generations may not be able to enjoy nature's services. We have developed a conceptual framework that establishes the links between biodiversity dynamics and abiotic change through time and space using artificial intelligence. Here, we apply this framework to a freshwater ecosystem with a known history of human impact and study 100 years of community-level biodiversity, climate change and chemical pollution trends. We apply explainable network models with multimodal learning to community-level functional biodiversity measured with multilocus metabarcoding, to establish correlations with biocides and climate change records. We observed that the freshwater community assemblage and functionality changed over time without returning to its original state, even if the lake partially recovered in recent times. Insecticides and fungicides, combined with extreme temperature events and precipitation, explained up to 90% of the functional biodiversity changes. The community-level biodiversity approach used here reliably explained freshwater ecosystem shifts. These shifts were not observed when using traditional quality indices (e.g. Trophic Diatom Index). Our study advocates the use of high-throughput systemic approaches on long-term trends over species-focused ecological surveys to identify the environmental factors that cause loss of biodiversity and disrupt ecosystem functions.

## eLife assessment

This **fundamental** study advances the analytic toolset and understanding of long-term series of biological (freshwater) communities, and the impact of humans on these. The authors highlight the value of including not only spatiotemporal scales in biodiversity assessments but also some of the possible drivers of biodiversity loss. Analyzing their joint contribution as environmental stressors,

the authors provide **compelling** evidence that ecosystem assessment methods currently used by environmental regulators throughout Europe are not fit-for-purpose, and they identify several alternatives, more robust indicators of freshwater ecosystem health. The work is timely and will be of interest to ecologists, modelers and global warming scientists in general.

## Introduction

Biodiversity is the foundation of provisioning, regulating, supporting, and cultural ecosystem services (*Baert et al., 2016*), which underpin economic prosperity, social well-being and quality of life (*Cardinale et al., 2012*). Global biodiversity has been lost at an alarming rate in the past century, leading to what some have called the sixth mass extinction - biodiversity loss caused by human population growth and activities (*Naggs, 2017*). Biodiversity is threatened by agricultural land use, climate change, invasive species, pollution and unsustainable production and consumption (*Bonebrake et al., 2019*). Freshwater ecosystems have suffered the greatest biodiversity loss because of these anthropogenic drivers (*Ruckelshaus et al., 2020*). Experimental manipulation of biodiversity has demonstrated the causal links between biodiversity loss and loss of ecosystem functions (*Eisenhauer et al., 2019*). However, studies on multi trophic levels are scarce and largely focus on terrestrial and marine ecosystems; freshwater ecosystems, especially lakes and ponds, are not well represented in multitrophic experimental manipulations, (*Dornelas et al., 2018*). These holistic studies are critical to understand the context-dependency of biodiversity-ecosystem functions relationships and to implement management measures to conserve biodiversity. However, a better understanding of the environmental factors with the largest impact on biodiversity, and their cumulative effect over time is urgently needed (*Halpern et al., 2015*).

Biodiversity action plans have been devised since the 1990s. However, most strategies have failed to stop or even reduce biodiversity decline (*Rounsevell et al., 2020*). This is because:

1. Biodiversity loss occurs at different spatial and temporal scales, and dynamic changes in community composition are the result of long-term ecological processes (*Eastwood et al., 2022*; *Nogués-Bravo et al., 2018*). State-of-the-art environmental and biological monitoring typically captures single snapshots in time of long-term ecological dynamics, failing to identify biodiversity shifts that may arise from cumulative impacts over time (*Eastwood et al., 2022*; *Nogués-Bravo et al., 2018*). Recent initiatives like BioTIME started collating databases with species presence and abundance recorded from time series across different ecosystems (*Dornelas et al., 2018*). However, freshwater ecosystems are poorly represented in these studies which at most encompass the last 10–25 years (*Blowes et al., 2019*). Although the large geographic breath of these studies is good to understand overall trends of biodiversity change, they are inadequate to identify drivers of biodiversity dynamics (*Halpern et al., 2015*; *Blowes et al., 2019*). Moreover, the taxonomic species assignment in these databases is oftentimes derived from traditional observational methods (e.g. microscopy), which cannot resolve cryptic diversity (*Blowes et al., 2019*). High cryptic diversity is common in freshwater invertebrates and primary producers, potentially impacting the assessment of biodiversity in these ecosystems (*Hirai et al., 2017*). More recently, *seda*DNA (environmental DNA extracted from sediment) has emerged as a promising tool to study decade-long biological dynamics (*Domaizon et al., 2017*). However, these studies focus on specific taxonomic groups [microbes (*Capo et al., 2019*) and ciliates (*Barouillet et al., 2022*)], failing to capture the community-level changes in any given ecosystem.

2. Biodiversity is threatened by multiple factors. Only by quantifying trajectories of abiotic, biotic, and functional systemic change over time, can we begin to identify the causes of biodiversity and ecosystem function loss (*Bonebrake et al., 2010*). Studies are emerging that investigate the impact of chemicals (*Groh et al., 2022*) or climate change (*Pecl et al., 2017*) on biodiversity. Yet, understanding the combined effect of these abiotic factors on biological communities is still challenging.

3. The lack of paired biological and abiotic long-term monitoring data is a limiting factor in establishing meaningful and achievable conservation goals. Even well-monitored species have time series spanning a few decades at best (*Halpern et al., 2015*; *Bonebrake et al., 2010*). Moreover, conservation efforts have historically focused on ecological surveys of few indicator species, the identification of which require specialist skills (e.g. light microscopy and taxonomy) and are low throughput (*Gillson and Marchant, 2014*). High-throughput system-level approaches providing

**eLife digest** Over long periods of time, environmental changes – such as chemical pollution and climate change – affect the diversity of organisms that live in an ecosystem, known as 'biodiversity'. Understanding the impact of these changes is challenging because they can happen slowly, their effect is only measurable after years, and historical records are limited. This can make it difficult to determine when specific changes happened, what might have driven them and what impact they might be having.

One way to measure changes in biodiversity over time is by analysing traces of DNA shed by organisms. Plants, animals, and bacteria living in lakes leave behind genetic material that gets trapped and buried in the sediment at the bottom of lakes. Similarly, biocides – substances used to kill or control populations of living organisms – that run-off into lakes leach into the sediment and can be measured years later. Therefore, this sediment holds a record of life and environmental impacts in the lake over past centuries.

Eastwood, Zhou et al. wanted to understand the relationship between environmental changes (such as the use of biocides and climate change) and shifts in lake biodiversity. To do so, the researchers studied a lake community that had experienced major environmental impacts over the last century (including nutrient pollution, chemical pollution and climate change), but which appeared to improve over the last few years of the 20th century.

Using machine learning to find connections over time between biodiversity and non-living environmental changes, Eastwood, Zhou et al. showed that, despite apparent recovery in water quality, the biodiversity of the lake was not restored to its original state. A combination of climate factors (such as rainfall levels and extreme temperatures) and biocide application (particularly insecticides and fungicides) explained up to 90% of the biodiversity changes that occurred in the lake. These changes had not been identified before using traditional techniques. The functional roles microorganisms played in the ecosystem (such as degradation and nitrogen metabolism) were also altered, suggesting that loss of biodiversity may lead to loss of ecosystem functions.

The findings described by Eastwood, Zhou et al. can be used by environmental regulators to identify species or ecosystems at risk from environmental change and prioritise them for intervention. The approach can also be used to identify which chemicals pose the greatest threat to biodiversity. Additionally, the use of environmental DNA from sediment can provide rich historical biodiversity data, which can be used to train artificial intelligence-based models to improve predictions of how ecosystems will respond to complex environmental changes.

biological, abiotic and functional changes over multiple decades are needed to understand links between biodiversity loss, drivers of changes and potential consequences on ecosystem functionality (*Eastwood et al., 2022*).

Recently, we have developed a conceptual framework that helps establish the links between biodiversity dynamics and abiotic environmental changes using artificial intelligence, examines emergent impacts on ecosystem functions, and forecasts the likely future of ecosystem services and their socio-economic impact under different pollution and climate scenarios (*Eastwood et al., 2022*). Here, we illustrate the first component of this framework in a freshwater ecosystem (Lake Ring, Denmark) with a well-documented human-impact over 100 years (*Cuenca Cambronero et al., 2018b*) by quantifying the interrelations between community-level functional biodiversity, biocides and climate (*Figure 1*). Historical records, supported by empirical evidence show that Lake Ring experienced semi-pristine conditions until the early 1940s (*Cuenca Cambronero et al., 2018a*). In the late 1950s, sewage inflow caused severe eutrophication. When the sewage inflow was diverted at the end of the 1970s, agricultural land use intensified, leading to substantial biocides leaching (*Cuenca Cambronero et al., 2018b*). The lake partially recovered from eutrophication and land use in modern times (>1999) but, as with every lake ecosystem in Europe, it experienced an increase in average temperature (*Cuenca Cambronero et al., 2018a*; *Cuenca Cambronero et al., 2018c*). We apply multilocus metabarcoding and mass spectrometry analysis to a dated sedimentary archive of Lake Ring. These data, complemented by biocides sale records and climate records, were studied with explainable network models with multimodal learning (*Baltrusaitis et al., 2019*) to identify drivers of functional biodiversity

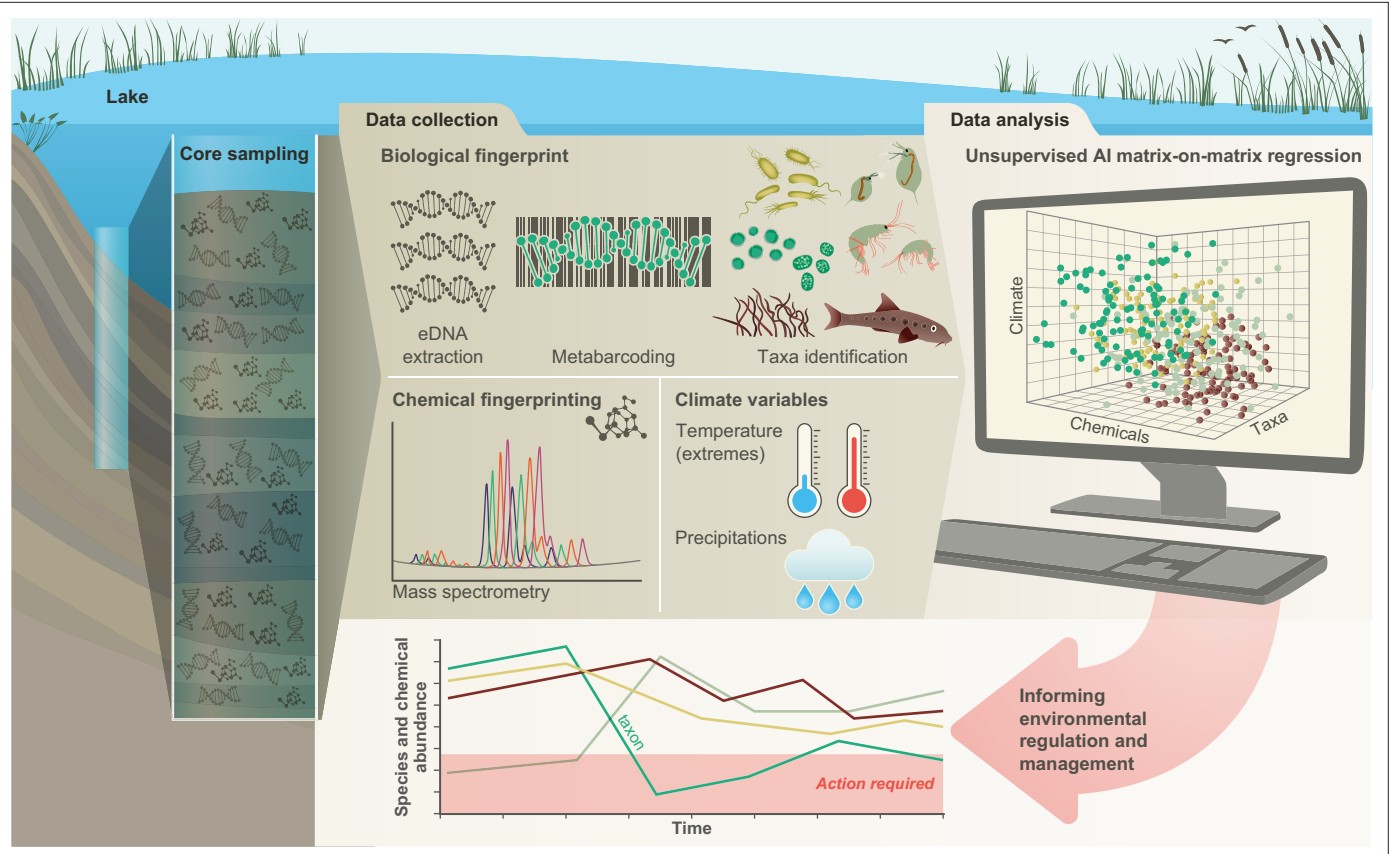

**Figure 1.** Conceptual framework. A sedimentary archive spanning 100 years was sampled from Lake Ring, Denmark and dated using radioisotopes. Both biotic and abiotic changes were empirically quantified through time: (1) community-level biodiversity was reconstructed by applying multilocus metabarcoding to environmental DNA isolated from layers of sediment (biological fingerprinting); (2) chemical signatures were quantified from the same sediment layers using mass spectrometry analysis (chemical fingerprinting); (3) climate data were collected from publicly available databases. Explainable network models with multimodal learning were applied to identify significant correlations between system-level biodiversity, chemical fingerprinting, and climate variables. Taxonomic units (families) impacted by environmental factors were identified and environmental factors ranked based on their effects on community biodiversity. This approach enables the prioritisation of conservation and mitigation interventions.

changes across major ecosystem shifts (*Figure 1*). The combination of explainable networks and multi-modal learning allow the simultaneous interrogation of data matrices describing different types of data. A symmetric matrix-on-matrix regression is typically used to identify the components that covary within a matrix (e.g. environmental variables), and among matrices (e.g. environmental variables and eDNA taxonomic units). Given the well-documented human-impact over time, Lake Ring represents an excellent natural system to demonstrate the power of systemic approaches in biological and functional monitoring.

## Results

### Freshwater community dynamics across 100 years

A sedimentary archive was collected from Lake Ring in November 2016 with a gravity corer; the core was sliced in 34 layers of 0.5 cm, which corresponded to a temporal resolution of about 3 years per layer across 100 years. This estimate was based on a radiometric chronology of the core completed in 2018 (see Materials and methods). Lake Ring has a well-known and documented history of human impact over the past century. The lake transitioned over time from a semi-pristine environment to eutrophication, and later to high pesticide pollution due to intensification of agricultural land-use in the area surrounding the lake. In modern times (>1999), the lake partially recovered (see Materials and methods for more details) (*Cuenca Cambronero et al., 2018b*). Hereafter, we refer to the lake transitions across these statuses as lake phases.

**Table 1.** PERMANOVA on beta diversity.

Permutational Multivariate Analysis of Variance using weighted Unifrac distances ASV matrices testing for pairwise differences between lake phases across the five barcodes used in the study (16SV1, 16SV4, 18S, COI, rbcL) with 999 permutations. Significant terms (*P*-values <0.05 after applying Benjamini & Hochberg correction for multiple testing) are in bold. The lake phases are as follows: SP - semi-pristine; E - Eutrophic; P - pesticides; R - recovery.

| Phase | | 16SV1 | | 16SV4 | | 18S | | COI | | rbcL | |
|---|---|---|---|---|---|---|---|---|---|---|---|
| 1 | 2 | R2 | p adj | R2 | p adj | R2 | p adj | R2 | p adj | R2 | p adj |
| SP | E | 0.4349 | **0.0067** | 0.5533 | **0.0017** | 0.2968 | **0.0033** | 0.0432 | 0.705 | 0.2879 | 0.0914 |
| SP | P | 0.6290 | **0.0025** | 0.8515 | **0.0017** | 0.4459 | **0.0033** | 0.3868 | **0.0033** | 0.3920 | **0.0125** |
| SP | R | 0.6956 | **0.0025** | 0.9026 | **0.0017** | 0.3841 | **0.0033** | 0.3178 | **0.0033** | 0.5084 | **0.0033** |
| E | P | 0.3959 | **0.006** | 0.7399 | **0.0017** | 0.1249 | 0.15 | 0.3198 | **0.005** | 0.1555 | 0.1511 |
| E | R | 0.5656 | **0.0025** | 0.8520 | **0.0017** | 0.1816 | **0.0075** | 0.2806 | **0.0033** | 0.6019 | **0.0033** |
| P | R | 0.3026 | **0.0025** | 0.3724 | **0.0017** | 0.1029 | 0.15 | 0.1924 | **0.012** | 0.3605 | **0.0033** |

We quantified community-level biodiversity over a century (1916–2016) by applying high-throughput multilocus metabarcoding (18S, 16SV1, 16SV4, COI and rbcL barcodes) to bulk environmental DNA (eDNA) extracted from layers of a dated sedimentary archive from Lake Ring. After denoising, the number of unique ASVs and total number of reads across all samples (including median number of reads per sample) found per barcode were as follows: 18S - 2,023 ASVs, 569,761 total reads (median 12,893 reads); 16SV1 - 4,022 ASVs, 842,619 total reads (median 20,798 reads); 16SV4 - 5,270 ASVs, 552,064 total reads (median 13,816 reads); COI - 822 ASVs, 362,616 total reads (median 9,595 reads); rbcL - 417 ASVs, 366,489 total reads (median 9,443 reads). Alpha diversity did not significantly vary across the lake phases for both prokaryotes and eukaryotes (*Appendix 1—figure 1*) and was proportionally higher in the prokaryotic (16 S barcodes) than in the eukaryotic community (18S barcode). Conversely, the invertebrate community (COI barcode), and the diatom community (rbcL barcode), showed significant changes over time across the lake phases, reflecting taxon-specific patterns over time (*Appendix 1—figure 1*). Even though the alpha diversity varied over time, it was not consistently lower in historical than modern communities across the barcodes, allowing us to exclude bias in the preservation state of environmental DNA.

The community composition (beta diversity) changed significantly in the transition between lake phases (*Table 1*; *Figure 2A*; *Appendix 1—figure 2*). The overall eukaryotic community composition changed over time across all lake phases (*Table 1*; *Figure 2A*; 18S). However, the composition of the primary producers (e.g. rbcL) changed significantly only in the transition between the pesticide and the eutrophic phases, whereas the invertebrate's community (e.g. COI) changed significantly only between the pesticide and the recovery phases (*Table 1*; *Figure 2A*; rbcL, COI). The significant changes in community composition identified by the PERMANOVA analysis were driven by two families of primary producers [*Chlorophyceae* (green algae), *Mediophyceae* (diatoms)] and seven families of invertebrates, [Monhysterida (nematode worms), Oligohymenophorea (ciliates), Calanoida (zooplankton), Ploimida (rotifers), Chaetonotida (gastrotrichs), *Thoracosphaeraceae* (dinoflagellates) and Calanoida (copepods)] (*Figure 2B*; 18S). In the transition from the semi-pristine to the eutrophic phase, the relative abundance of rotifers and green algae declined in favour of calanoids and diatoms, respectively (*Figure 2B*; 18S). The proportion of diatoms, worms and nematodes increased in the transition from the eutrophic to the pesticide phase, while the proportion of calanoids and gastrotricha declined (*Figure 2B*; 18S). The taxonomic composition of the recovery phase showed a relative increase in ciliates and gastrotricha as compared to the pesticide environment (*Figure 2B*; 18S). *Vampirellidae* (Vampire amoebae feeding on algae) were relatively more abundant in the eutrophic than in the other phases, in which primary producers were also more abundant (*Figure 2B*, 18S). The composition of the recovery and semi-pristine phases differed significantly, suggesting an incomplete recovery of the lake over time to this date (*Table 1*; *Figure 2A*;18S).

The prokaryotic community significantly changed at each major transition between lake phases, consistently across the two barcodes (*Table 1*; 16SV1 and 16SV4). We observed two patterns in the

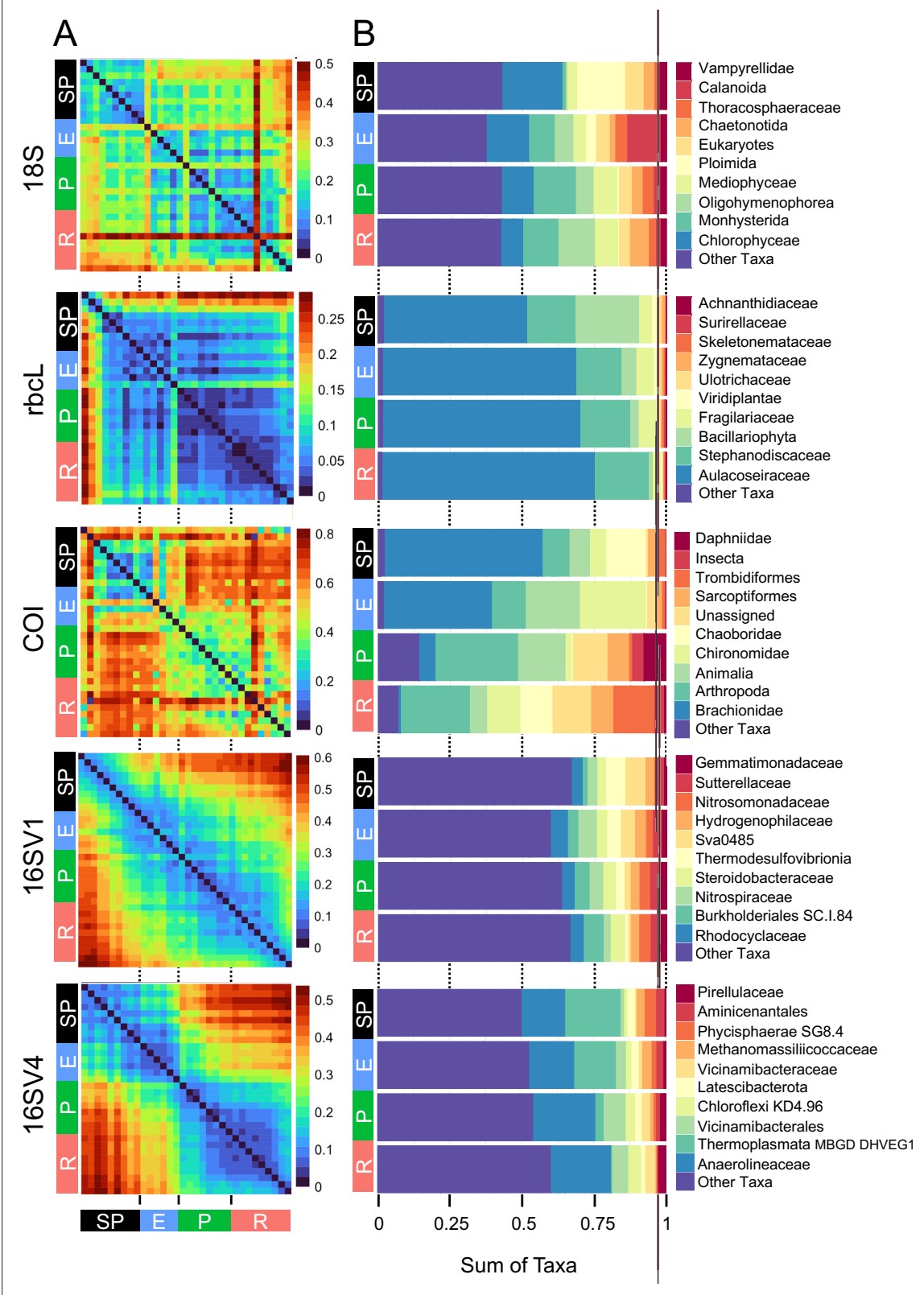

**Figure 2.** Biodiversity compositional changes. (**A**) Weighted unifrac beta diversity heatmaps between each pair of sediment layers spanning a century (1916–2016) for the five barcodes used in this study (18S, rbcL, COI, 16SV1 and 16SV4). The PERMANOVA statistics in *Table 1* support these plots. The scale used may be different among the heatmaps. (**B**) Taxonomic bar plots including the top 10 most abundant families identified across five barcodes (18S, rbcL, COI, 16SV1 and 16SV4). shown per lake phase: SP - semi-pristine; E - eutrophic; P - pesticides; R - recovery.

prokaryotic community composition over time: some taxonomic groups changed with the redox status of the sediment (e.g. acidophilus archaea [Thermoplasmata] and methanogenic archaea [*Methano-massiliicoccaceae*], which declined from the semi-pristine to the recovery phase [*Figure 2B*, 16SV4]); others changed over time consistently with the nutrient levels of the ecosystem. For example *Nitrospiraceae* (nitrite oxidizers) were more abundant in high-nutrient environments (eutrophic and pesticides) than in lower nutrient environments (semi-pristine and recovery) (*Figure 2B*; 16SV1).

Changes in the invertebrate community were driven by *Brachionideae* (rotifers) that were most abundant in the semi-pristine phase and declined over time; *Chironomidae* (lake flies) that were proportionally more abundant in the eutrophic and recovery phases and showed the lowest abundance in the pesticides phase; *Chaoboridae* (phantom midge larvae) that were only present in the semi-pristine and recovery phases; and *Daphniidae* (waterfleas) that were most abundant in the pesticide phase, but present throughout the 100 years of sampling (*Figure 2B*; COI). The diatom composition was stable over time, with only the semi-pristine phase having a more distinctive diatom assemblage profile dominated by *Bacillariophyta* (*Figure 2B*; rbcL). Diatoms are commonly used by regulators to derive the status of freshwater within the Water Framework Directive both for lakes and rivers (*Agency, 2020*). We used our rbcL data to derive a Lake Trophic Diatom Index (LTDI2) for Lake Ring following *Bennion et al., 2014*. This result confirmed our beta diversity analysis of non-significant changes over time of the diatom community (*Appendix 1—figure 3*).

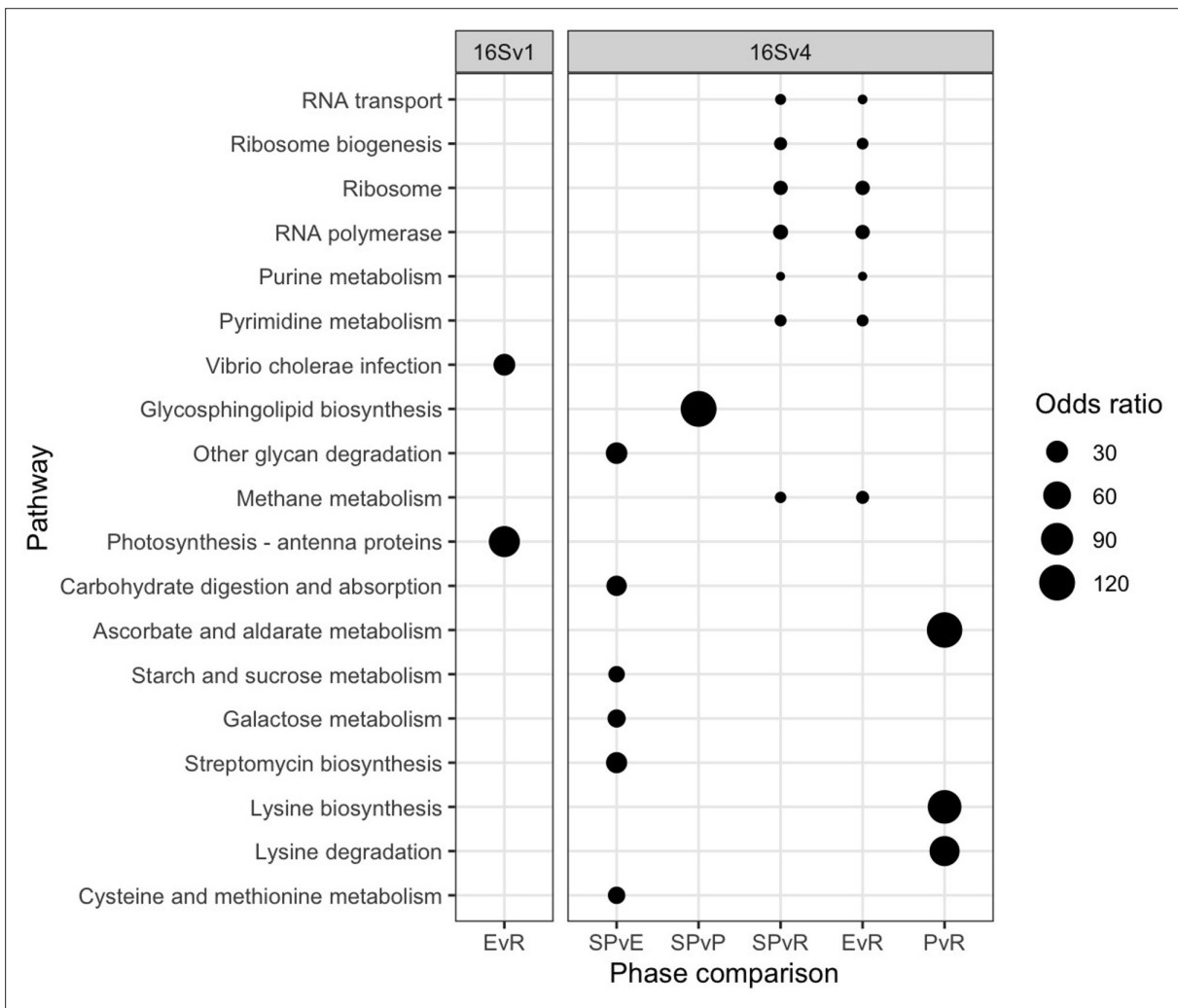

**Figure 3.** Functional analysis. Functional pathways that are significantly differentially enriched between lake phases are shown for the 16SV1 and the 16SV4 barcodes (Fisher's exact test, p <0.05). The lake phases are as in *Figure 2*: SP - semi-pristine; E - eutrophic; P - pesticides; R - recovery. Odds ratios indicate the representation of each pathway in the pairwise comparisons.

## Functional changes linked to community compositional shifts

Changes in freshwater community composition corresponded to significant shifts in the predicted functioning of the prokaryotic community (*Figure 3*). We predicted different functions between each pair of lake phases by identifying molecular functions enriched as quantified by functional orthologs (KO terms). A functional ortholog was defined from functions experimentally assigned to the prokaryotes identified with the 16 S rRNA in each lake phase. We predicted a total of 6,257 Kegg Orthologs (KO) terms from the 16SV1 and 6,828 from the 16Sv4 barcode across the lake phases. Of the total number of KO terms, 1,418 were significantly differentially abundant across the lake phases in the 16SV1 and 1,064 terms in the 16SV4 dataset, respectively. The functional KEGG pathways enriched within these KO terms and significantly differentially enriched between lake phases (Fisher's exact test, p <0.05) were 19 (17 for the 16SV4 and 2 for the 16SV1; *Figure 3*). Seven differentially enriched pathways were found between the semi-pristine and recovery phases and seven were found between the eutrophic and recovery phases (*Figure 3*; 16SV4). These pathways were linked to catabolic functions (purine and pyrimidine metabolism), RNA transport and biogenesis, fundamental for gene expression and protein folding. Six functional pathways were differentially enriched between the semi-pristine and the eutrophic phases that were linked to metabolism (including methane metabolism), degradation and biosynthesis (*Figure 3*; 16SV4). Three functional pathways that underpin carbohydrates metabolism, lysine biosynthesis and degradation were differentially enriched between the pesticide and recovery phases. The latter two functions are critical for mitochondrial function. A single pathway was differentially enriched between the semi-pristine and the pesticide phases, linked to lipid metabolism (glycosphingolipid biosynthesis; *Figure 3*; 16SV4). Two differentially enriched pathways were identified between the eutrophic and the recovery phases and underpin infection response and photosynthesis (*Figure 3*; 16SV1).

## Drivers of biodiversity change

To discover drivers of biodiversity change we applied sparse canonical correlation analysis (sCCA) to community biodiversity data and other parameters measured from Lake Ring, namely climate records collected from a weather station proximal to the lake, and sales records of biocides in Denmark between 1955 and 2015 from the Danish national archives. The biocide sales records proved to be a good representation of persistent chemicals in the lake sediment, as the quantification of the persistent halogenated pesticide DDT in the sliced sedimentary archive showed, by producing a very similar profile as the sales records over time (see methods section).

We discovered that insecticides and fungicides best explained changes in overall biodiversity, possessing the highest CCA loadings across the barcodes, followed by pesticides and herbicides (*Supplementary file 1A*). Among the climate variables, yearly minimum temperature explained the largest biodiversity changes, whereas other climate variables had a variable impact across the barcodes and hence taxonomic groups (*Supplementary file 1B*).

Having ranked biocides and climate variables that best explained changes in overall biodiversity, we identified correlations between taxonomic groups (assigned at family level where possible) and individual abiotic variables. Correlations were identified between a total of 44 eukaryotic (18S, COI and rbcL) families and abiotic variables; of these correlations, 32 were with biocides and 33 with climate variables (some correlations involved the same taxonomic group correlating with multiple environmental factors). Of the 32 families negatively correlated with biocides, the largest proportion co-varied significantly with insecticides (22 families - 69%) and fungicides (16 families - 50%), followed by herbicides (9 families - 28%) and pesticides (3 families - 9%) (*Supplementary file 2*). Of the 33 families correlated with climate variables, the largest proportion co-varied with summer precipitation (14 families - 42%); of these, 9 families were positively correlated and 5 were negatively correlated with summer precipitation. The next three highest proportions of families which co-varied with climate variables were summer atmospheric pressure (10 families - 30%; 7 positive and 3 negative correlations), mean minimum temperature (9 families – 27%; 6 positive and 3 negative correlations) and highest recorded temperature (8 families – 24%; 7 positive and 1 negative correlations) (*Supplementary file 2*).

The number of unique prokaryote families significantly negatively correlated with biocides was 99, 19 of which were identified by both 16 S barcodes. Following from the sCCA analysis, significant negative correlations were observed between 60 (60.6%) families and insecticides, followed by 59 families

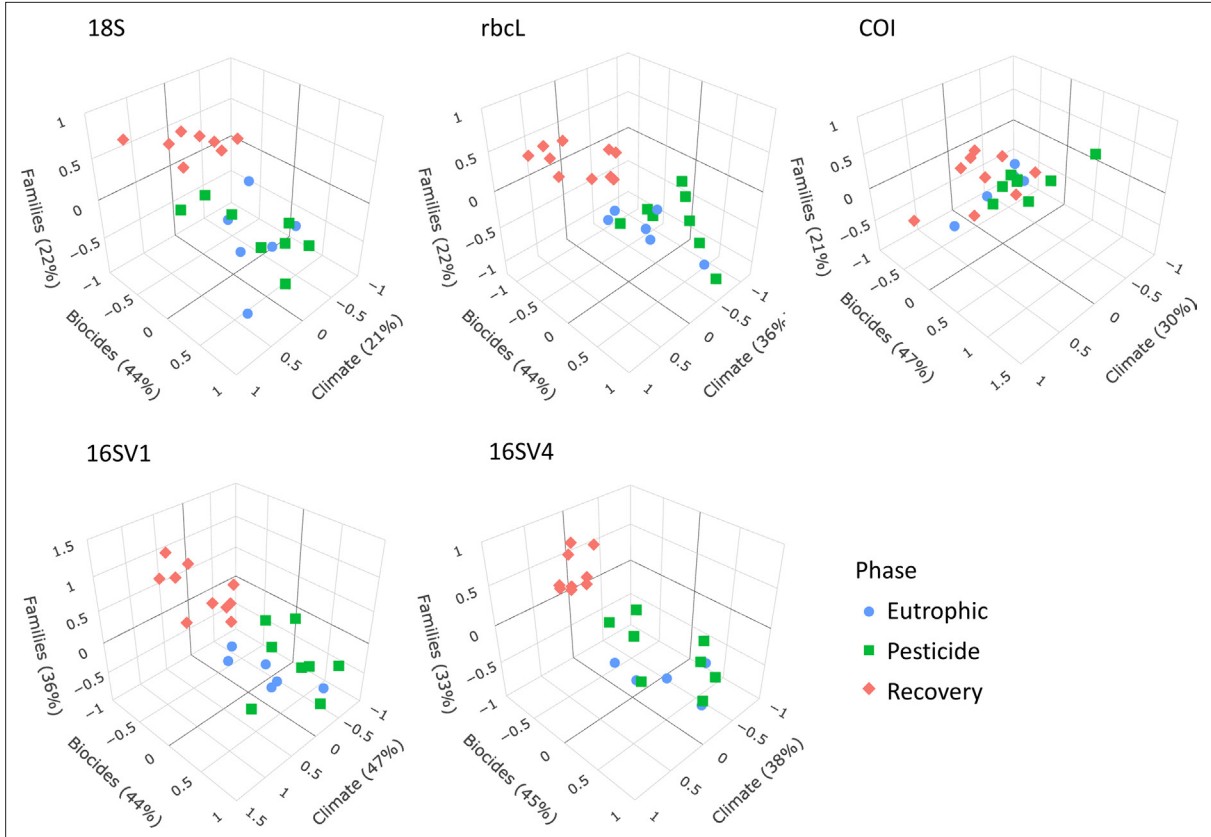

**Figure 4.** sCCA 3D plots. Sparse canonical correlation analysis 3D plots for the five barcodes used (18S, rbcL, COI, 16SV1 and 16SV4), showing the proportion of biodiversity variance explained by the biocides and climate variables. As biocides were introduced around the 1960s, this analysis spans the most recent three lake phases (Eutrophic, Pesticide and Recovery). Interactive version available: https://environmental-omics-group.github.io/Biodiversity_Monitoring/.

and fungicides (59.6%), 40 families and herbicides (40.4%), and 25 families and pesticides (25.3%) (*Supplementary file 2*; overall). A total of 105 non-redundant correlations were identified between prokaryotic families and climate variables, 6 of which were found in both 16 S barcodes. Of the total families correlating with climate variables, 69 (65.7%) significantly correlated with mean minimum temperature. Of these, 38 were positive and 31 were negative correlations. Thirty-five families (33.3%) significantly correlated with summer precipitation; of these, 11 were positively and 23 were negatively correlated. Twenty-nine families (27.6%) significantly correlated with the lowest recorded temperature; of these 20 were positive and 9 were negative correlations. Twenty-six families (24.8%) significantly correlated with mean summer temperature; of these 13 were positively and 13 negatively correlated. Twenty-three families (21.9%) significantly correlated with maximum daily precipitation; of these, 3 were positively and 20 were negatively correlated. Eleven families (10.4%) significantly correlated with highest recorded temperature; of these 3 were positively and 8 were negatively correlated (*Supplementary file 2*).

We applied sCCA to identify families that correlated both with climate variables and biocides (*Figure 4*). As biocides were introduced only in 1960, only the most recent three lake phases were included in this analysis. The eukaryotic biodiversity compositional change was predominantly explained by biocides (*Figure 4*; 18S; Biocides: 44%), followed by climate variables (*Figure 4*; 18S; climate variables: 22%). Up to 22% of the diatoms compositional change was explained by biocides (44%) and climate variables (36%). However, the abiotic variables only separated the recovery from the other two lake phases (*Figure 4*), supporting significant biodiversity compositional shifts observed in the beta diversity analysis (*Figure 2A*; *Table 1*). Similarly, the invertebrate community compositional changes were explained prevalently by biocides (47%), followed by climate variables (30%), which only separated the recovery phase from the other two lake phases. Climate and biocides almost

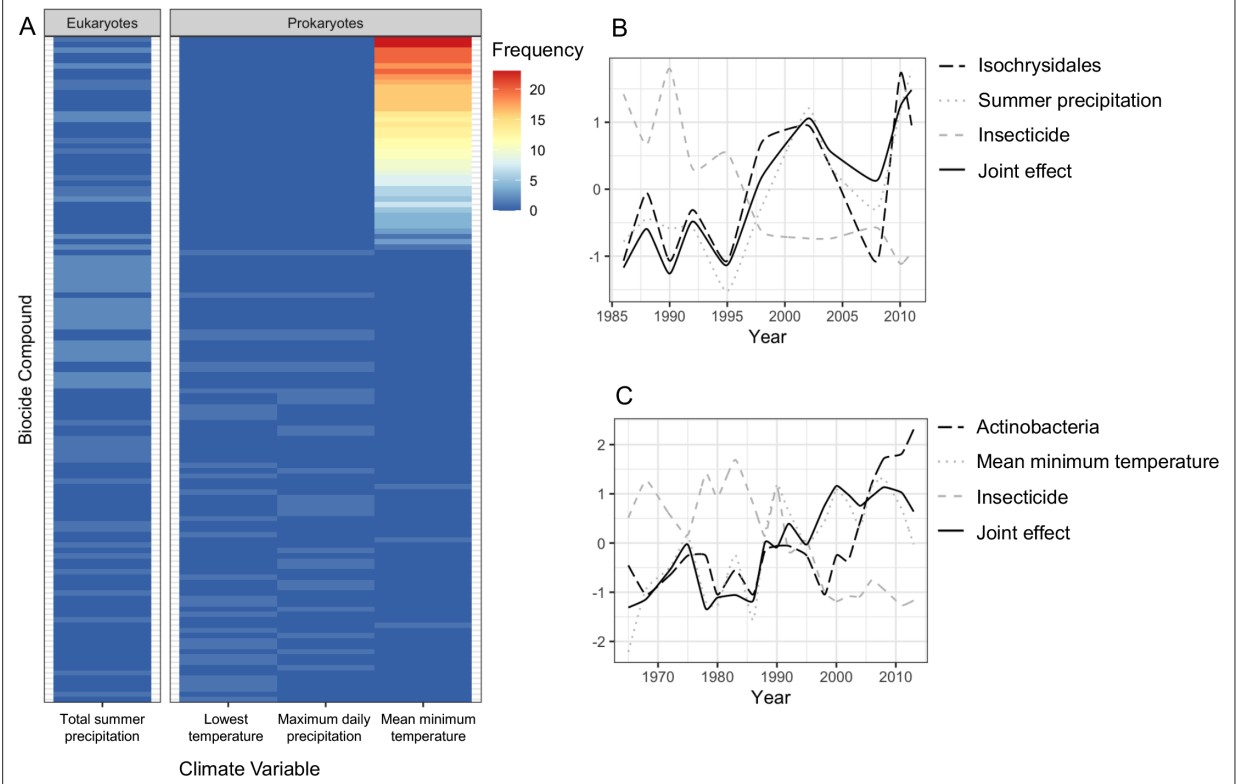

**Figure 5.** Joint effects of environmental variables on biodiversity. (**A**) heatmap showing the frequency of joint effects of biocides and climate variables in eukaryotes (data from the 18S barcode) and prokaryotes (combined data from 16SV1 and 16SV4 barcodes). The biocides are ranked based on their correlation coefficient with taxonomic units and climate variables. Ranking of biocide types is provided in *Supplementary file 3*; (**B**) temporal correlation between the family Isochrysidales, summer precipitation and insecticides. The joint effect of summer precipitation and insecticides is also shown; (**C**) temporal correlation between Pleosporales, insecticides and mean minimum temperature. The joint effect of insecticides and mean minimum temperature is also shown. The families' relative abundance over time in plots B and C are standardized values.

equally explained up to 36% of the prokaryote biodiversity compositional change across the lake phases (16SV1 - biocides: 44%, climate variables 47%; 16SV4 - biocides 45%, climate variable 38%). Following from this analysis, joint effects of biocides and climate variables were observed for 23 prokaryotic (16 S) and two eukaryotic (18S) families (*Figure 5A*), whereas no joint effects were identified on the diatom (rbcL) and the invertebrate (COI) communities (*Figure 5A*; *Supplementary file 3*). The most frequent joint effects on prokaryotes were from insecticides and mean minimum temperature combined (*Figure 5A*; *Supplementary file 3*). Joint effects between herbicides and maximum daily precipitation or between herbicides and lowest recorded temperature were rare (*Figure 5A*; *Supplementary file 3*). The joint effects on the eukaryotic community were observed between insecticides and summer precipitation (*Figure 5A*; *Supplementary file 3*).

The biocide types showing joint effects with environmental variables were ranked based on their correlation coefficient over time (*Supplementary file 3*). The top ranked insecticides most frequently showing these joint effects with climate variables and an adverse effect on both prokaryotes and eukaryotes were: oxydemeton-methyl (organothiophosphate insecticide, primarily used to control aphids, mites, and thrips), mevinphos (organophosphate insecticide used to control insects in a wide range of crops) and dicofol (organochlorine acaricide chemically related to DDT). Additionally, parathion (organophosphate insecticide and acaricide), carbaryl (1-naphthyl methylcarbamate used chiefly as an insecticide), dieldrin (organochlorine insecticide, developed in alternative to DDT) and thiometon (organic thiophosphate insecticide) showed adverse effects with only the prokaryotic community. Examples of joint effects on specific families are shown in *Figure 5B and C*. The temporal dynamics of Isochrysidales, a coccolith-producing microalgae, was affected by the joint effect of summer precipitation and insecticides (*Figure 5B*), whereas the temporal dynamics of the PeM15 group of Actinobacteria was affected by the joint effect of insecticides and mean minimum temperature (*Figure 5C*).

## Discussion

### Continuous long-term biomonitoring from a pristine baseline

State-of-the-art paleoecological monitoring typically uses direct observations (light microscopy) of species remains to assess the ecological status of freshwater ecosystems. These approaches are low throughput and require specialist skills (*Moraitis et al., 2019*). Direct observations are inherently biased towards species that leave fossil remains; species identification is strongly reliant on well-preserved remains in environmental matrices; and cryptic species diversity cannot be resolved (*Hirai et al., 2017*). Recently, automated acquisition of microfossil data using artificial intelligence has been proposed as an alternative to human inspection for reconstructing long-term biological changes (*Itaki et al., 2020*). However, this approach relies on the completeness of reference databases and of the fossil remains, suffering from the same limitations of direct observations minus the low throughput aspects. Efforts to catalogue temporal changes in biodiversity have recently started to understand changes in species richness and assemblages in different geographic regions of the globe (*Blowes et al., 2019*). These efforts are important to understand the extent of overall biodiversity loss. However, there are only a handful of existing datasets that span more than 50 years and many of the multidecal biodiversity time series are limited to terrestrial and marine ecosystem, with freshwater ecosystems being marginally represented (*Blowes et al., 2019*). Moreover, long-term freshwater studies tend to focus on indicator species or specific taxonomic groups (e.g. invertebrates), rather than capturing community-level patterns (*Dornelas et al., 2018*). Developments in the field of *seda*DNA have addressed the limitations of direct observations, utilising the properties of eDNA (*Capo et al., 2019*). However, *seda*DNA studies have predominantly focused on microorganisms as proxies for ecosystems' health [e.g. cyanobacteria (*Picard et al., 2022*); ciliates (*Barouillet et al., 2022*); parasitic taxa (*Talas et al., 2021*)], with other taxonomic groups less well represented. Our study addresses some of the challenges of direct observations as it is not reliant on fossil remains. However, the completeness of the community taxonomic assignment depends on the completeness of reference databases. We acknowledge that our taxonomic classification may be incomplete.

Whereas the application of high-throughput sequencing technologies requires training, these technologies are well established with publicly available standard operating procedures. As compared to direct observations, high-throughput sequencing provides replicable results regardless of the operator. Moreover, the application of metabarcoding to *seda*DNA or more generally eDNA can be outsourced to established environmental services, removing the need for training if it is a limiting factor.

Studies of temporal dynamics typically start from an already shifted baseline and rely on discrete observations (*Barouillet et al., 2022*). Our study alleviates these limitations by providing a continuous community-level analysis of biological changes over recent evolutionary times and starting from a relatively undisturbed environment. However, eDNA-based studies suffer from limitations linked to the level of preservation of nucleotides in environmental matrices. Although it has been shown that DNA can be recovered from lacustrine and marine sediments as far back as the Holocene (*Slon et al., 2022*), biases might still exist due to the degradation of eDNA, especially over geological times (*Jia et al., 2022*) and in warmer climates (*Mauvisseau et al., 2022*). In addition, physio-chemical changes in sediment and soil may affect the assemblage and composition of prokaryotic communities that can survive in extreme conditions, including anoxic environments. However, it has been shown that slightly alkaline water (pH 7–9) facilitates DNA preservation (*Jia et al., 2022*). Whereas we cannot exclude that the eDNA in our study suffers from some of the mentioned biases, we expect DNA degradation not to have affected our study significantly. This is because we observed non-significant difference in species richness over time in both the prokaryotic (16 S barcode) and eukaryotic (18S barcode) communities. DNA degradation would have resulted in lower alpha diversity with increasing age of the sediment. Preservation of DNA in our study is also favoured by the time frame studied (100 years as opposed to millennia), the stable pH since the 1960s (data prior to 1960s were not recorded), and the latitude of Lake Ring which experiences average yearly temperatures below 15°C. All these factors are known to reduce microbial activity, allowing a better preservation of DNA in sediment (*Giguet-Covex et al., 2014*).

Although the overall species richness did not change significantly over time, species assemblages significantly changed over time. Non-significant changes in alpha diversity coupled with significant changes in beta diversity over time have been reported for existing time series of biodiversity data

in marine and terrestrial environments, even if the length of the time series rarely exceeded four decades (*Blowes et al., 2019*).

## Insecticides and extreme temperatures drive changes in functional biodiversity

Threats to biodiversity pose a significant challenge because they change over time and may result in additive adverse effects (*Bonebrake et al., 2019*). Long-term continuous observations are preferable to short-term observations because they can reveal correlations and possible causation between biological changes and abiotic drivers of change (*Gillson and Marchant, 2014*). Using eDNA-based data on multitrophic biodiversity over the past 100 years, we identified the taxonomic groups within the prokaryotic and eukaryotic communities that significantly contributed to community assemblages shifts. Whereas the prokaryotic community was overall changing at each major transition between lake phases, changes in the eukaryotic community were driven by different taxonomic groups in the transition between lake phases. The diatom community, typically used by regulators as an indicator of freshwater ecological status, did not change significantly over time, as the beta diversity and the LTDI2 index revealed. These results strongly suggest that a system-level approach, like the one proposed here, may be more appropriate than species or taxon-specific approaches. Our approach showed that diatom communities are not a reliable representation of the ecological status of freshwater ecosystems and are not good indicators of environmental change. Our approach provides a major advantage over traditional direct observations by identifying both taxonomic and functional changes of freshwater biodiversity in a high-throughput fashion. The analysis of temporal trends of biodiversity from a semi-pristine baseline through impacted environments provides a new reference point for regulators to define biodiversity in semi-pristine conditions.

Even if Lake Ring partially recovered from eutrophication and biocide pollution in modern times, both the contemporary eukaryotic and prokaryotic communities are significantly different from the semi-pristine historical community, as the PERMANOVA on beta diversity demonstrates. Our findings align with other studies using *seda*DNA on decennial timeframes focusing on prokaryotes (e.g. cyanobacteria *Cao et al., 2020*), whereas studies on eukaryotic compositional changes are just emerging to enable quantitative comparative assessments (*Zhang et al., 2021*). Studies on prokaryotic and eukaryotic assemblages based on short experimental manipulations suggest that natural communities can return to their original state before a perturbation occurs (*Hillebrand and Kunze, 2020*). However, longer term experimental manipulations show a different perspective with irreversible changes in biodiversity composition and function (*Fordham, 2015*). These long(er)-term experimental manipulations and our study suggest that empirical observation of multi trophic changes over time in natural systems are critical to understand the context-dependency of biodiversity-environmental impact relationships and assess the resilience of natural ecosystems.

Changes in community assemblages are important because they can be associated with changes in functional biodiversity. Although biodiversity variables include taxonomic, phylogenetic, and functional attributes, most studies have focused on generic taxonomic diversity measures – usually measured as species richness or abundance, ignoring functional biodiversity (*Li et al., 2020*). Biomass and changes in biomass only capture productivity, while disregarding other metrics, such as decomposition or resource turnover (*Gounand et al., 2018*). A complete assessment of biodiversity should include functionality (*Eisenhauer et al., 2019*). In particular, enzyme activities are relevant because they exhibit the functions encoded in genes and reflect the role of microbiota in the transfer of matter and energy from low to high trophic levels in ecosystems. Changes in biological assemblages over time and across lake phases in our study resulted in significant changes in functional biodiversity, observable through changes in metabolic, biosynthesis and degradation functions of the prokaryotic community demonstrated by differentially abundant KEGG pathways between lake phases. Catabolic functions, metabolism (including methane metabolism), degradation and biosynthesis were differentially enriched between the recovery and other lake phases. These are key functions for the survival of organisms. For example, change is metabolic potency and the ability to break down complex molecules into smaller ones (catabolism and degradation) may affect survival and fitness of living organisms by influencing the uptake of nutrients.

Predicting the functional profiles of prokaryotic communities based on their taxonomic composition has its limitations. Predictions of functions linked to human gut microbes tend to be more accurate

than predictions on other communities because reference databases are developed on currently available genomes, which are biased towards microorganisms associated with human health and biotechnology (*Choi et al., 2017*). Because of the bias in reference databases, functional predictions may be more accurate for basic metabolic and housekeeping functions (essential cellular functions that are evolutionary conserved), which are more commonly annotated (*Mi et al., 2019*). Therefore, it is possible that we underestimated the predicted changes in functional biodiversity driven by environmental change in our study. Yet, we were able to detect important functional changes (e.g. metabolism and biosynthesis essential for survival) in correspondence with major ecosystem shifts (e.g. from semi-pristine to recovery phase).

In recent years, an increasing number of studies have documented impacts on biodiversity driven by climate change (*Pecl et al., 2017*), whereas chemicals are thought to pose a negligible threat to biodiversity because living organisms can adapt and evolve (*Groh et al., 2022*). Adaptation to environmental change can happen, but it comes at a cost that can reduce resilience of natural populations to multiple stressors or novel stress (*Cuenca-Cambronero et al., 2021*). Our study showed that chemicals and climate variables each explain up to 47% of biodiversity compositional changes and that the joint effect of insecticides/fungicides and yearly extreme temperature/summer precipitation best explained changes in overall biodiversity. The joint effects of insecticides and extreme temperature events affected prokaryotes by altering their functionality and changing their metabolic, biosynthesis and degradation functions. The joint effect of insecticides and summer precipitation best explained changes in primary producers and grazers. This result aligns with previous studies showing that the effect of chemicals on freshwater can be exacerbated by temperature/precipitation, because of changes in the bioavailability, adsorption, elimination and relative toxicity of chemicals by water organisms (*Pinheiro et al., 2021*). Higher temperatures increase diffusion of chemical molecules, resulting in faster uptake by living organisms and hence toxicity (*Patra et al., 2015*). In some cases, higher temperatures result in effects on the organism's metabolic ability to reduce a chemical's toxicity. Our study hints at examples of both mechanisms, distinguishing between families that are negatively and positively correlated with climate variables.

The resolution and reliability of our data-driven systemic approach goes beyond the current state-of-the-art, enabling us to identify the specific abiotic factors, down to the commercial name of biocides, that in isolation or combined with climate variables affected specific families of prokaryotes and eukaryotes. Our algorithm provides a high degree of confidence that surpasses state-of-the-art analysis, which predominantly identify patterns of co-occurrence of taxa within communities e.g. Correlation-Centric Network approach (*Yang et al., 2020*). A step in the right direction to capture complex interactions between biotic and abiotic variables is the network analysis of co-occurrence patterns among physico-chemical and biological variables using random forest machine learning algorithms (e.g. *Tse et al., 2018*). This approach is hypothesis-free and allows the identification of synchronicity between various environmental variables and *seda*DNA sequence variation. However, even when applied to temporal trends, it does not quantify joint effects of environmental factors on biodiversity. So far, random forest machine learning algorithms have only been applied to prokaryotic communities, disregarding other taxonomic groups and providing a partial understanding of community-level patterns and responses (*Tse et al., 2018*).

A potential limitation of our approach is that correlations identified in field surveys do not demonstrate causation. However, they generate testable hypotheses that can be proven experimentally in controlled mesocosm experiments as explained in *Eastwood et al., 2022*, providing a potentially transformative approach.

## Implications for conservation and management of biodiversity

Some of the greatest challenges in biodiversity conservation faced by water resource managers is the limited information available on a time scale sufficient to assess long-term changes of aquatic ecosystems. Large-scale models that link environmental drivers to biological indicators are lacking (*Solimini et al., 2005*), even if some countries have tried to introduce semi-quantitative indices to assess the ecological status of freshwater (*Archaimbault and Dumont, 2010*). Regulators must rely on approaches ingrained into environmental law, even though they have been proven inadequate (e.g. TDI), as the continuous decline in biodiversity demonstrates (*Pecl et al., 2017*). Even when direct links between biological indicators and abiotic drivers can be established, these rely on indicator species

(e.g. a fish, an alga and an invertebrate) used as proxies for ecosystem health (*Kanno, 2016*). Our data-driven approach provides a novel way to address regulatory needs. However, the use of data-driven, systemic approaches requires critical changes in current environmental practice and a shift to whole-system evidence-based approaches. The transition to the novel methodologies proposed here will require changes in regulatory frameworks, following a test and acceptance phase, as well as a buy-in from regulators. Our study is a proof of concept that the drivers of biodiversity loss can be identified with higher accuracy than currently possible, generating hypotheses that can be tested experimentally. Our data-driven approach enabled us to identify insecticides and temperature as strong drivers of biodiversity loss, both in prokaryotes and eukaryotes. The confirmation of these findings across multiple freshwater ecosystems has the potential to inform conservation and mitigation interventions, leading to an improved preservation of functional biodiversity.

## Materials and methods
### Environmental and paleoecological profile of Lake Ring

Lake Ring is a shallow mixed lake in Jutland, Denmark (55°57′51.83″ N, 9°35′46.87″ E) with a well-known history of human impact (*Cuenca Cambronero et al., 2018b*). A sedimentary archive was collected from Lake Ring in November 2016 with an HTH-type gravity corer; the core was sliced in 34 layers of 0.5 cm and stored in dark and cold (–20 °C) conditions. A radiometric chronology of this sediment was completed in 2018 by Goldsmith Ecology Ltd following standard protocols (*Appleby, 2001*), and provided an accurate dating of the sediment to the year 1916. According to this chronology the core covered 100 years at a resolution of ca. 3 years intervals. To reduce potential contamination when handling older sediment layers each layer of sediment was handled in a PCR-free and DNA-free environment. Dating of sediment was conducted by direct gamma assay, using ORTEC HPGe GWL series well-type coaxial low background intrinsic germanium detector. Sediment samples with known radionuclide profiles were used for calibration following *Appleby, 2001*.

We used, historical records, direct chemical analysis of sediment, and physico-chemical records to reconstruct the paleoecological environment of Lake Ring. According to historical records, the lake was semi-pristine until the 1940s. In the late 1950s, sewage inflow from a nearby town increased nutrient levels resulting in eutrophication. The sewage inflow was diverted at the end of the 1970s, but this period coincided with agricultural land-use intensification (>1980), causing biocides leaching into the lake. The lake partially recovered in modern times (>1999), experiencing a partial return to its original trophic state and reduced impact from biocides (*Cuenca Cambronero et al., 2018b*).

Physico-chemical variables were measured in the lake between 1970 and 2016, even though data are sparse and discontinuous, limiting their use in a machine learning or statistical framework. To complement the historical records, we obtained climate data from the Danish Meteorological Institute (*Supplementary file 4*). The climate data were collected from a weather station 80 km from Lake Ring. Air and water surface temperature typically have a positive correlation for shallow streams and lakes (*Livingstone and Lotter, 1998*; *Preudhomme and Stefan, 1992*). Hence, we used the data from the weather station as an estimate of the lake water temperature. We also observed a tight correlation between the recorded water temperature in Lake Ring and the summer air temperature recorded by the weather station. In addition, we procured sales records of biocides in Denmark between 1955 and 2015 from the Danish national archives (*Supplementary file 4*). To assess whether the biocide sales records were a good representation of persistent chemicals in the lake sediment, we quantified the persistent halogenated pesticide DDT in the sliced sedimentary archive of Lake Ring, applying gas chromatography with mass spectrometry analysis. Sediment samples were lyophilized and freeze dried in a lyophilizator using a Christ Beta 1–8 LSCplus freeze-dryer, (Martin Christ GmbH, Osterode am Harz, Germany), to avoid analyte loss during water removal. Following lyophilization, the sediment samples were sieved through 0.4 mm meshes and homogenised. Approximately 1 g of dry sediment was weighed into pre-cleaned glass tubes and spiked with 100 ng of deuterated [2H8- 4,4`- DDT], used as an internal (surrogate) standard, followed by 1 g of copper powder (Merck, Dorset, UK) for sulphur removal. The sediment samples were extracted using 5 ml of hexane: acetone (3:1 v/v), vortexed for 5 min, followed by ultrasonication for 15 min and centrifugation for 3 min at 5000 rpm. The supernatant was transferred to a clean, dry tube and the process was repeated twice for each sample. The combined extract was then evaporated to dryness under a gentle stream of N2 and reconstituted in

2 mL of hexane. Sulphuric acid (3 ml) was used to wash the reconstituted crude extract. The organic phase was allowed to separate on top of the acid layer then transferred to another clean dry test tube. The remaining acid layer was washed twice, each with 2 ml of Hexane. The combined clean extract and washes was evaporated under a gentle stream of Nitrogen, reconstituted into 150 µl of iso-octane containing 100 pg/µl of PCB 131 used as syringe (recovery) standard. Quantification of target DDTs was conducted on a TRACE 1310 GC coupled to an ISQ single quadrupole mass spectrometer (Thermo Fisher Scientific, Austin, TX, USA) operated in electron ionization (EI) mode according to a previously reported method (*Wong et al., 2009*).

## Biodiversity fingerprinting across 100 years

### eDNA extraction and metabarcoding sequencing

We applied multilocus metabarcoding or marker gene sequencing to environmental DNA (eDNA) extracted from the 34 layers of sediment from the biological archive of Lake Ring using a laminar flow hood in a PCR-free environment to obtain a fine-grained temporal quantification of taxonomic diversity and relative abundance of taxonomic groups. eDNA was extracted from the dated sediment layers - *seda*DNA - using the DNeasy PowerSoil kit (Qiagen), following the manufacturer's instructions. Negative aerial and PCR controls were used; in addition, positive controls for PCR consisting of duplicates of three random samples from the sedimentary archive, were used. The duplicated samples were very similar, providing confidence in the approach used (*Appendix 1—figure 2*). Triplicates of each *seda*DNA sample were amplified with a suite of five nuclear and mitochondrial PCR primers (barcodes) to capture presence and relative abundance of eukaryotes (18S) (*Hadziavdic et al., 2014*), macroinvertebrates (COI) (*Leray et al., 2013*), primary producers (focus on diatoms; rbcL) (*Zimmermann et al., 2014*), and prokaryotes (16SV1 and 16SV4) (*Caporaso et al., 2011*) using Q5 HS High-Fidelity Master Mix (New England Biolabs) and following the manufacturer's instructions. A negative control in triplicate per plate was used. Paired end 250 bp amplicon libraries were obtained using a two-step PCR protocol with 96x96 dual tag barcoding to facilitate multiplexing and to reduce cross-talk between samples in downstream analyses (*MacConaill et al., 2018*) by EnviSion, BioSequencing and BioComputing at the University of Birmingham (https://www.envision-service.com/). PCR1 and PCR2 primers, as well as annealing temperatures per primer pair in PCR1 are in *Supplementary file 5*. Excess primer dimers and dinucleotides from PCR1 were removed using Thermostable alkaline phosphatase (Promega) and Exonuclease I (New England Biolabs). PCR2 amplicons were purified using High Prep PCR magnetic beads (Auto Q Biosciences) and quantitated using a 200 pro plate reader (TECAN) using qubit dsDNA HS solution (Invitrogen). A standard curve was created by running standards of known concentration on each plate against which sample concentration was determined. PCR2 amplicons were mixed in equimolar quantities (at a final concentration of 12 pmol) using a biomek FXp liquid handling robot (Beckman Coulter). The final molarity of the pools was confirmed using a HS D1000 tapestation screentape (Agilent) prior to 250 bp paired-end sequencing on an Illumina MiSeq platform.

### Bioinformatics

The reads were demultiplexed using the forward PCR1 primer sequence and cutadapt 3.7.4 with an error rate of 0.07, equating to one allowed mismatch. The quality of sequences was assessed with FASTQC (*Wingett and Andrews, 2018*) and multiqc (*Ewels et al., 2016*). Sequences were then imported into QIIME2 v 2021.2 (*Bolyen et al., 2019*), trimmed, filtered, merged and denoised using the QIIME2 DADA2 module (*Callahan et al., 2016*) using default parameters and trimming low quality sections and reverse primer [forward read 0–10 trimmed front, 214–225 truncation; reverse read 17–26 trimmed front, 223–247 truncation]. After denoising, the following samples had zero reads remaining: 16SV1, 16SV4, rbcL, and COI-negative PCR controls; COI aerial negatives A and B; 16SV1 sampleID 8. The taxonomic assignment was completed with the naive-bayes taxonomic classifiers trained using different reference databases, depending on the barcode: the SILVA v138 database was used for the assignment of the 16SV1, 16SV4, and 18S reads (*Yilmaz et al., 2014*); the diat. barcode v9.2 was used for the assignment of rbcL reads (*Rimet et al., 2019*); and the Barcode of Life Database was used for the COI reads (*Robeson et al., 2021*). The taxonomy was assigned using qiime feature-classifier classify-sklearn and used at family level where possible (*Pedregosa, 2011*). When classification was not possible at family level, the lowest classification possible was used. The

taxonomic barplots were plotted per barcode using ggplot2 v3.3.5 (*Wickham, 2016*) in R v4.0.2 (*Anderson, 2001*) and including the top 10 most abundant families. All other taxa were collapsed in the plots under 'other taxa'.

All samples were rarefied (16SV1 at 10,250 reads; 16SV4 at 10,400 reads; 18S at 9,070 reads; COI at 3,580 reads; rbcL at 4,650 reads) to achieve normalisation for calculating Alpha and Beta diversity metrics with QIIME2 (*Bolyen et al., 2019*). The following samples did not meet the rarefaction cutoff: 16SV1: aerial negatives A, B, C; 16SV4: aerial negatives A, B, C and sampleID 62 sample;18S: aerial negatives A,B,C, negative PCR control, sampleID 18, positive control replicate 62; rbcL: aerial negative A, B, and sampleIDs 50, 54, 60; COI sampleIDs 40, 64. Alpha diversity differences among lake phases, using shannon entropy, were tested with Kruskal-Wallis test and beta diversity differences among lake phases, calculated as weighted unifrac distances, were established with a PERMANOVA test (*Kolde, 2019*). Alpha diversity was plotted using ggplot2 v3.3.5 with R v4.0.2. Heatmaps of weighted unifrac Beta diversity between each pair of sediment layers were plotted with the pheatmap v1.0.12 in R v4.0.2 (*Douglas et al., 2020*).

The function of the microbial communities across the four lake phases were predicted with PICRUST2 (*Mandal et al., 2015*) plugin in QIIME2 (*Bolyen et al., 2019*), using the rarefied reads. Differentially abundant KEGG Orthology (KO) terms between pairs of lake phases were identified using the ANCOM plugin (*Lin et al., 2013*) in QIIME2 (*Bolyen et al., 2019*) and were mapped onto KEGG pathways with enriched pathways identified using a Fisher Exact test.

## Drivers of biodiversity change

To identify correlations between biological assemblages (families identified through the *seda*DNA sequencing) and drivers of change, we focused on biocides and climate variables, using sparse Canonical Correlation Analysis (sCCA; it can be thought of as consensus PCA on multiple data matrices) followed by Sliding Window (Pearson) Correlation (SWC) analysis (*Appendix 1—figure 5*). Physicochemical variables were not used in this analysis because of their sparsity (data rarely met the Sliding Window correlation criteria of 5 continuous values) and low variation over time (*Appendix 1—figure 4A*). sCCA is a tool for integrating and discovering complex, group-wise patterns among high-dimensional datasets (*Parkhomenko et al., 2009*). While most forms of machine learning require large sample sizes, sCCA uses fewer observations to identify the most correlated components among data matrices and captures the multivariate variability of the most important features (*Nakagawa and Cuthill, 2007*).

Matrices consisting of rarefied ASV reads per barcode, climate data and biocide types were used as input in the analytical pipeline summarised in . After the sCCA analysis the ASVs were assigned to family level where possible or at the next lowest classifier. The first step of the pipeline is preparing input matrices for ASVs, climate variables and biocides (*Appendix 1—figure 5*; Step 1). The following step is a matrix-on-matrix regression, applied to correlate families called from the ASVs with either biocide type or climate variables (*Appendix 1—figure 5*; Step 2). The top five components of the correlations, based on loading values, that explained the largest covariance between matrices were extracted from the sCCA, and the abiotic factors (climate variable and biocide type, separately) ranked according to their contribution to the overall covariance. A Sliding Window (Pearson) Correlation (SWC) analysis followed this step and was applied to each pair of vectors represented by the top ranked abiotic factor and the families. This approach was used to identify abiotic factors (either climate variables or biocide types) that significantly correlated with families over time, using the criterion that their Pearson correlation coefficient should be larger than 0.5, i.e. large effect size (*Buckland and Gey, 1994*) with an FDR adjusted p-value (padj) <0.05 following 10,000 permutations (*Appendix 1—figure 5*; Step 3). The minimum sliding window size was set to 5 time points, corresponding to 15% of the total time window for which families, biocides and climate data were available (the 34 sediment layers from the sedimentary archive span 100 years). Time intervals with more than 50% zero values in either the biotic or the abiotic data were discarded from downstream analyses to reduce false positives. A recall rate was used to quantify the number of ASVs within a family that were individually significantly correlated with the abiotic variables over all ASVs in a given family (*Buckland and Gey, 1994*). The families that co-varied with either biocide types or climate variables over time were retained if they showed a Pearson correlation coefficient >0.5, a padj <0.05 and a

recall rate >0.5 (90% quantile of the recall rates of all families; *Appendix 1—figure 5*; Step 4). This conservative approach enabled us to reduce noise from spurious correlations and improve accuracy.

The combined effect of environmental factors may have an augmented impact on biodiversity. To identify the combined effect of climate variables and biocides on the lake community biodiversity, we applied again the sCCA analysis (*Appendix 1—figure 5*; Step 5). For this analysis, we selected the climate variables and biocide types contributing the largest covariances in the correlation analysis in Step 4. Their combined effect on a family was considered to be significant if the biocide type and the climate variable were each significantly correlated with the family over the same time window, and their average Pearson correlation was >0.5 with padj <0.05 (SWC analysis with 10,000 permutations) (*Appendix 1—figure 5*; Step 6). The biocide type and the climate variable were interpreted to have an joint effect on a given family if the linear combination of the biocide type and the climate variable had a larger Pearson correlation coefficient than each of the correlations between the family and the biocide type and the family and the climate variable individually, in the same time interval with padj <0.05 (with 10,000 permutations in the SWC analysis).

Within each biocide type that significantly correlated with a family, we established their ranking based on the correlation coefficient (*Appendix 1—figure 5*; Step 6). Significant Pearson correlations that identified the joint effect of climate variables and individual biocides on a given family were identified with the same criteria outlined above (Pearson correlation >0.5; padj <0.05; SWC with 10,000 permutations). Chemicals with more than 50% null values or Pearson correlation coefficients <0.5 were discarded.

## Acknowledgements

We thank Kerry Walsh and Glenn Watts, the UK Environment Agency, for helpful discussions on the application of the approach presented here within regulatory frameworks. The metabarcoding data were generated by EnviSion, BioSeqencing and BioComputing at the University of Birmingham (https://www.envision-service.com/). The DDT chemical data were generated by the GEES Mass Spectrometry Facility at the University of Birmingham. Sediment sampling and dating was completed by Goldsmith Ecology, Somerset. We thank Stephen Kissane for technical assistance in generating high throughput sequencing data, Dr Xiaojing Li for helpful discussions on functional analysis and Chantal Jackson for the artwork of *Figure 1*. This work was funded by the Alan Turing Institute (under EPSRC grant R-BIR-001); and the NERC highlights grant LOFRESH (NE/N005716/1). NE is supported by the Biotechnology and Biological Sciences Research Council (Midlands Integrative Biosciences Training Partnership (MIBTP; BB/M01116X/1)). LO and HH have been supported by the RobustNature Cluster of Excellence Initiative (internal prefunding of Goethe University Frankfurt).

## Additional information

### Funding

| Funder | Grant reference number | Author |
| --- | --- | --- |
| Alan Turing Institute | R-BIR-001 | Luisa Orsini |
| Natural Environment Research Council | NE/N005716/1 | Luisa Orsini |
| Goethe-Universität Frankfurt am Main | RobustNature Cluster of Excellence Initiative | Luisa Orsini Henner Hollert |
| Biotechnology and Biological Sciences Research Council | BB/M01116X/1 | Niamh Eastwood |

The funders had no role in study design, data collection and interpretation, or the decision to submit the work for publication.

## Author contributions
Niamh Eastwood, Formal analysis, Funding acquisition, Investigation, Visualization, Writing – original draft, Writing – review and editing; Jiarui Zhou, Software, Formal analysis, Visualization, Methodology, Writing – original draft, Writing – review and editing; Romain Derelle, Formal analysis, Writing – review and editing; Mohamed Abou-Elwafa Abdallah, William A Stubbings, Investigation, Methodology, Writing – review and editing; Yunlu Jia, Sarah E Crawford, Henner Hollert, Methodology, Writing – review and editing; Thomas A Davidson, Holly Bik, Resources, Writing – review and editing; John K Colbourne, Simon Creer, Funding acquisition, Writing – review and editing; Luisa Orsini, Conceptualization, Supervision, Funding acquisition, Writing – original draft, Project administration, Writing – review and editing

## Author ORCIDs
Niamh Eastwood ⓘ https://orcid.org/0000-0003-2969-6091
Jiarui Zhou ⓘ http://orcid.org/0000-0002-1025-718X
Mohamed Abou-Elwafa Abdallah ⓘ http://orcid.org/0000-0002-4624-4073
William A Stubbings ⓘ http://orcid.org/0000-0002-8538-4693
Thomas A Davidson ⓘ http://orcid.org/0000-0003-2326-1564
Simon Creer ⓘ http://orcid.org/0000-0003-3124-3550
Luisa Orsini ⓘ https://orcid.org/0000-0002-1716-5624

Joint Public Review https://doi.org/10.7554/eLife.86576.3.sa1
Author Response https://doi.org/10.7554/eLife.86576.3.sa2

## Additional files

### Supplementary files
• Supplementary file 1. sCCA analysis. CCA loadings calculated with sparse canonical correlation analysis for biocides (A) and climate variables (B). The categories of biocides are insecticides, fungicides, pesticides and herbicides. The environmental variables are mean minimum temperature, maximum daily precipitation, highest recorded temperature, mean summer temperature, summer precipitation, annual total precipitation, summer atmospheric pressure and lowest recorded temperature.

• Supplementary file 2. Correlations between biodiversity and environmental variables. Summary of correlations between taxonomic units identified through the five barcodes (18S, 16SV1, 16SV4, rbc, and COI) and environmental variables, including biocides and climate factors. The taxonomic name and the number of significant correlations between a taxonomic unit and environmental variables, is followed by a correlation value, associated p-adjusted value and recall rate for each variable. The taxonomic units are reported at the lowest taxonomic assignment possible (f – family; o – order; c- class; p – phylum; null - unassigned). Results are collated per barcode, each in a separate tab. The last tab lists only taxonomic units that significantly correlated with the environmental variables based on the combined criteria of Pearson correlation value greater than 0.5, adjusted P-value smaller than 0.05 and recall rate greater than 0.5 along with the direction of the correlation.

• Supplementary file 3. Joint effects between biocides and climate variable. The biocides showing significant joint effect with climate variables are ranked based on their correlation coefficient. The barcode and identified families that are affected by the joint effect of a climate variable and biocides type are shown. The order in which the biocide types are ranked is the same used to plot *Figure 5*.

• Supplementary file 4. Lake Ring metadata. Dating record for Lake Ring, climate data collected from a weather station adjacent to the lake, and sales records for biocides are shown. The year of sampling (year), the sample ID, the depth of the sediment layer measured in centimetres (Depth), climate variables (annual mean temperature °C, summer mean temperature °C, mean minimum temperature °C, mean maximum temperature °C, highest recorded temperature °C, lowest recorded temperature °C, mean atmospheric pressure hPa, summer mean atmospheric presure hPa, annual total precipitation mm, summer precipitation mm, maximum daily precipitation mm, No. of days with snow cover, annual mean cloud cover, and summer mean cloud cover) and record of biocides sales between the 1950s and 2016 in tonnes/year and separated per class (insecticides, herbicides, fungicides and pesticides).

• Supplementary file 5. PCR primers. (1) PCR1 primers with bibliographic references, expected

fragment size (bp), annealing temperature (°C) and primer sequences (in black) with overhang to prime the sequencing flow cell; (2) PCR2 primers consisting of Nextera adapters, universal tail and overhang sequence.

• MDAR checklist

### Data availability

The metabarcoding sequences generated for this project are available at BioSample ID SAMN22315717 and SAMN22315798. Code used to process and analyse the data in this study are available at GitHub (copy archived at *Environmental-Omics-Group, 2022*).

The following datasets were generated:

| Author(s) | Year | Dataset title | Dataset URL | Database and Identifier |
|---|---|---|---|---|
| Niamh E, Jiarui Z, Romain D, Abou-Elwafa MA, William AS, Yunlu J, Sarah EC, Thomas AD, John KC, Simon C, Holly B, Henner H, Luisa O | 2021 | sample 10 - 18S COI rbcL | https://www.ncbi.nlm.nih.gov/biosample/SAMN22315717 | NCBI BioSample, SAMN22315717 |
| Niamh E, Jiarui Z, Romain D, Abou-Elwafa MA, William AS, Yunlu J, Sarah EC, Thomas AD, John KC, Simon C, Holly B, Henner H, Luisa O | 2021 | Negative Control - 16S | https://www.ncbi.nlm.nih.gov/biosample/SAMN22315798 | NCBI BioSample, SAMN22315798 |

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

# Appendix 1

The information contained in this appendix serves to describe the metabarcoding data and provide the reader with quality of data. We present the alpha diversity captured by each barcode used in this study across the 100 years covered by the study (*Appendix 1—figure 1*). The biological samples used as PCR positive controls showed small variance, providing confidence across the lake phases (*Appendix 1—figure 2*).

The diversity captured by the rbcL barcode was compared to the diversity expressed by the Trophic Diatom Index, which is a standard reference to determine water quality according to the Water Framework Directive guidelines. The diversity captured by the Trophic Diatom Index shows low variation over time (*Appendix 1—figure 3*) and is discussed in the main text alongside the diversity captured by the rbcL barcode.

Lake Ring physio-chemical parameters over time are shown in *Appendix 1—figure 4A*, followed by a reporting of biocides (Tons/year) from the 1960 to modern times (*Appendix 1—figure 4B*). The distribution of fungicides, herbicides, insecticides, pesticides and multi-targeted biocides are shown over a period of 50 years in Denmark. To assess whether the sales record was a good proxy for the biocides in the lake studied here, we compared the sales record and the empirical quantification of DDT from the sediment layers (*Appendix 1—figure 4C*). The two patterns show a very good overlap, suggesting that sales records are a good proxy for persistent chemicals in sediment.

The AI pipeline used in this study to identify the likely drivers of biodiversity change over time in Lake Ring is described in *Appendix 1—figure 5*. The main steps used in the pipeline to correlate biodiversity dynamics, biocides and climate variables is shown. In *Supplementary file 1*, sCCA analysis identifies the chemical class and climate variables loading for each barcode used, showing that insecticides are likely the main drivers of community biodiversity changes, followed by fungicides. Furthermore, the minimum temperature and precipitation were identified as strong drivers of biodiversity changes among the climate variables studies. An in-depth analysis of the environmental factors and taxonomic units identified with the metabarcoding analysis, enabled us to establish correlations between taxonomic units and specific environmental variables, to be used for further experimental validation to confirm causality (*Supplementary file 2*). The joint effect of climate variable and biocide classes can be found in *Supplementary file 3*. This analysis allowed us to identify the combined adverse effect of the two categories of climate variables on specific taxonomic units. For transparency we report the metadata used in the analyses presented in our study, including physio-chemical parameters measured in Lake Ring over time, as well as the biocides sales records for Denmark (*Supplementary file 4*). For completeness, the PCR primers used in the two-step PCR described in our work are in *Supplementary file 5*.

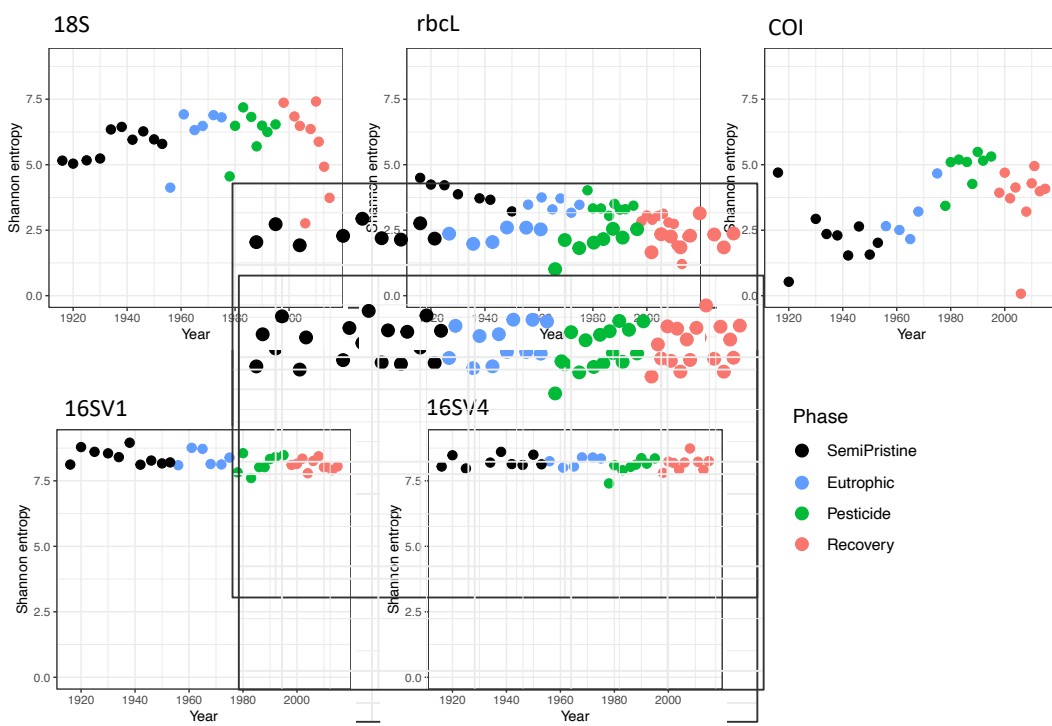

**Appendix 1—figure 1.** Alpha diversity. Alpha diversity, measured as Shannon entropy, is shown for the five barcodes used in this study (16SV1, 16SV4, 18S, COI and rbcl) between 1916-2016. The four lake phases are colour-coded as follows: Black - Semi-pristine; blue - Eutrophic; green - Pesticides; red - Recovery. Kruskal-Wallis test across all phases: 18S: h 4.199, Pval = 0.241; rbcL: h 21.677, Pval<0.000; COI: h 16.958, Pval = 0.001; 16SV1: h 7.001, Pval = 0.072; 16SV4: h 2.220, Pval = 0.528.

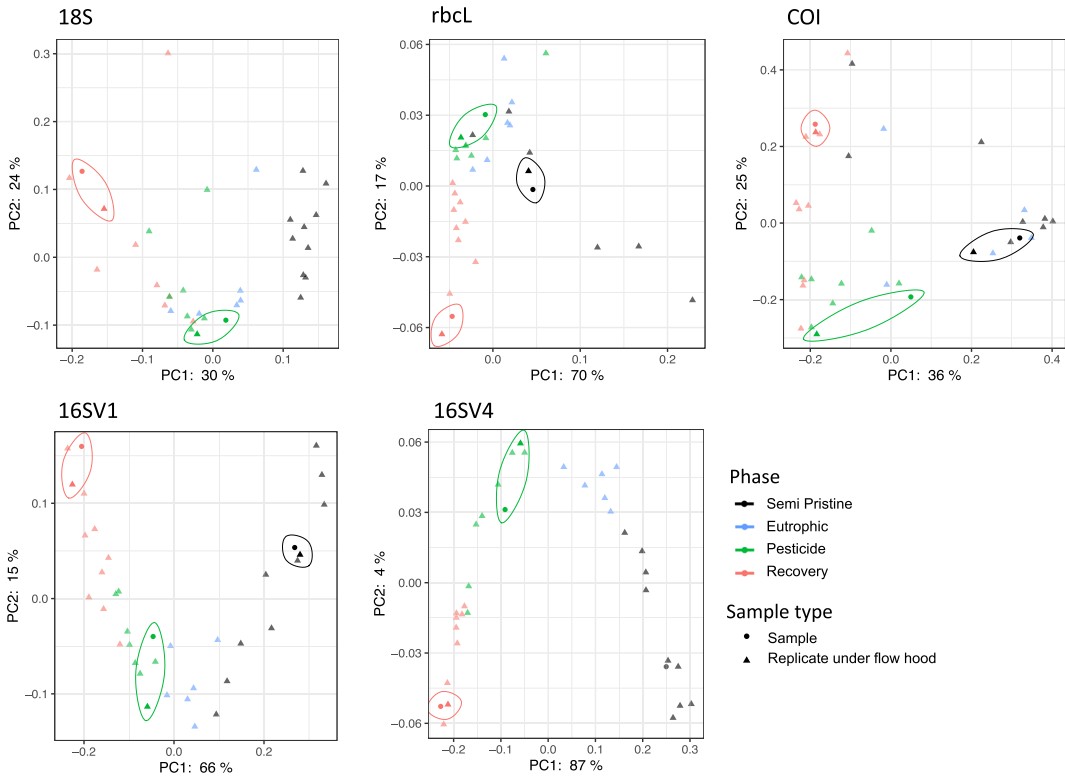

**Appendix 1—figure 2.** Principal Coordinate Analysis. PCoA visualization of weighted unifrac distance between samples. Positive controls for PCR consist of duplicates of up to three samples from the sedimentary archive for each of the five barcodes used in the study (16SV1, 16SV4, 18S, rbcL, and COI). Replicated samples are circled. The four lake phases are colour-coded as follows: Black - Semi-pristine; blue - Eutrophic; green - Pesticides; red - Recovery.

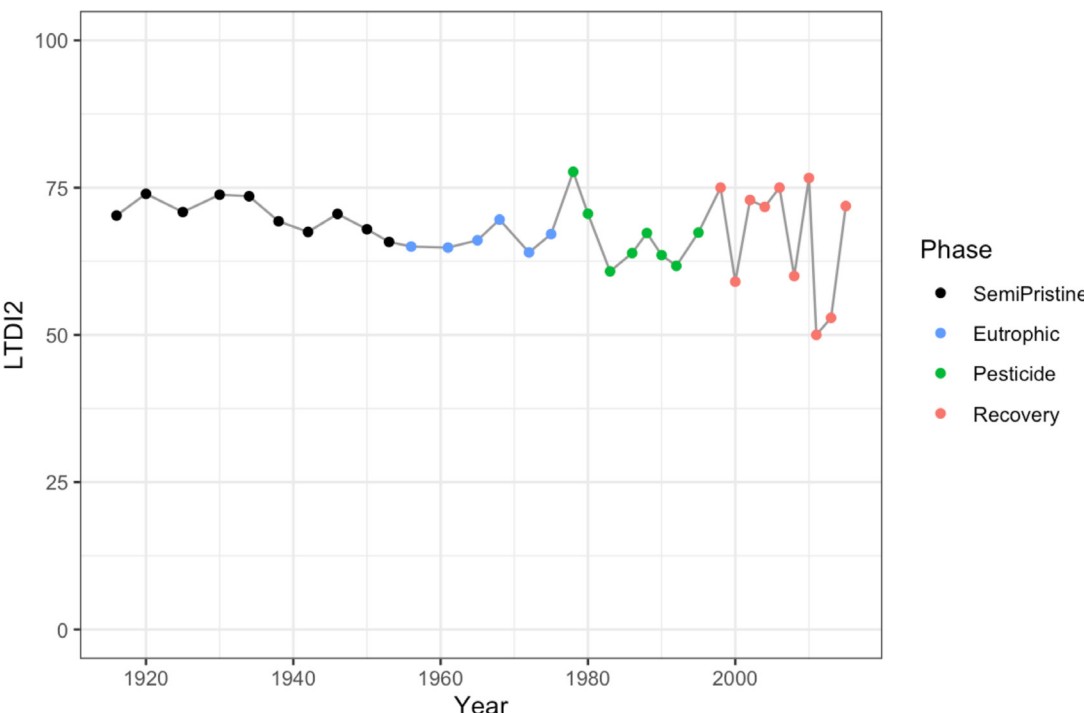

**Appendix 1—figure 3.** Trophic Diatom Index. LTDI2 calculated using the diatom species identified in our study between 1915 and 2015 with the rbcL barcode and the 'DARLEQ3' (Diatoms for Assessing River and Lake Ecological Quality) tool. Mean value of 67.59, standard deviation 6.3. The four lake phases are colour-coded as follows: Black - Semi-pristine; blue - Eutrophic; green - Pesticides; red - Recovery.

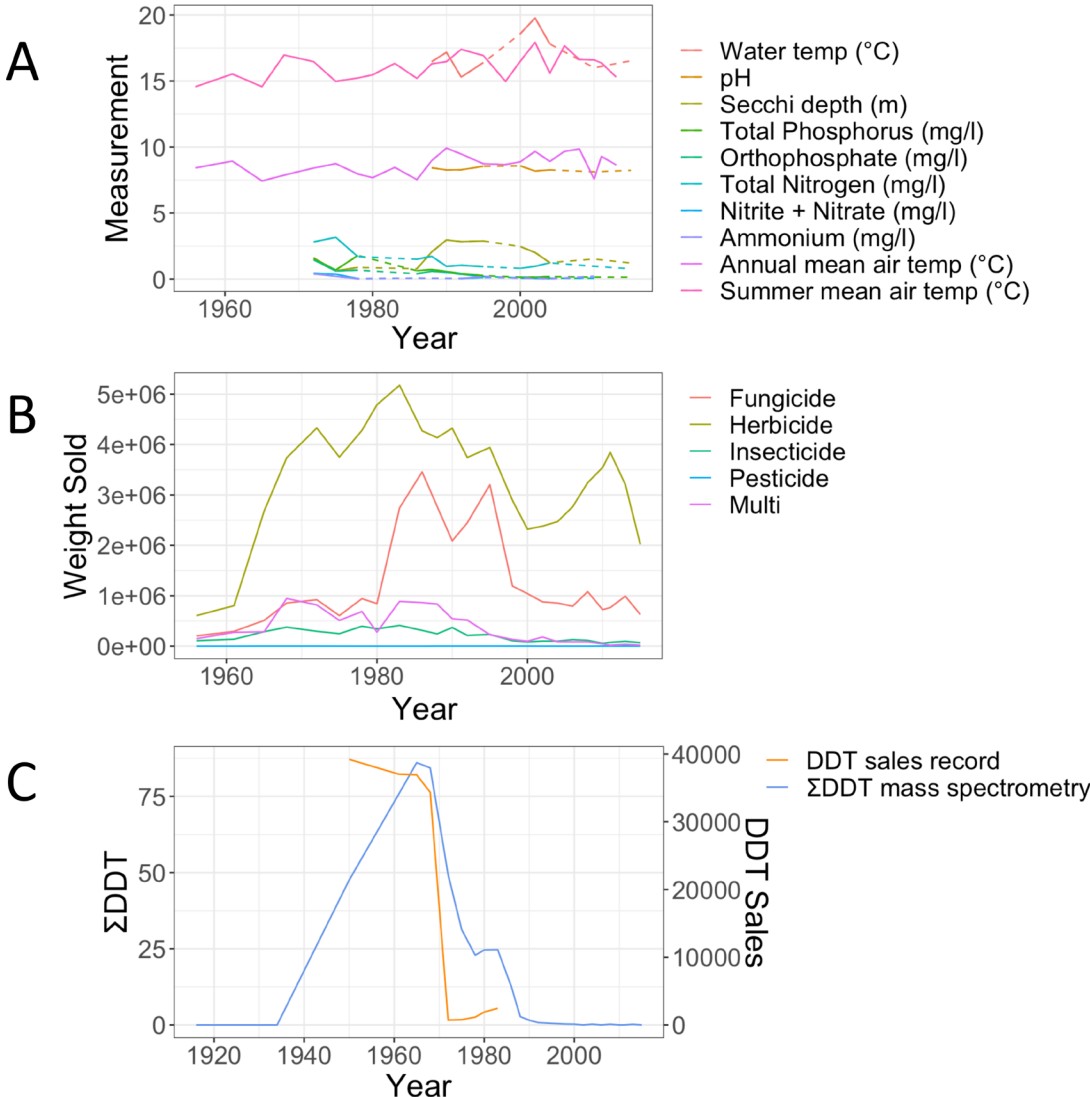

**Appendix 1—figure 4.** Biocides records. (**A**) Records of physico-chemical parameters measured in Lake Ring. Dotted lines indicate missing data points. Summer and annual mean temperature were recorded at a weather station 80km from Lake Ring. (**B**) Record of biocides sales in Denmark (Million Tons/Year) between 1950 and 2016, downloaded from the Danish national archives; (**C**) empirical record of DDT measured from the sediment layers of Lake Ring using mass spectrometry analysis (ng/g; blue) and plotted against the sales record in Denmark (Million Tons/year; orange). DDT was banned in Denmark in 1986.

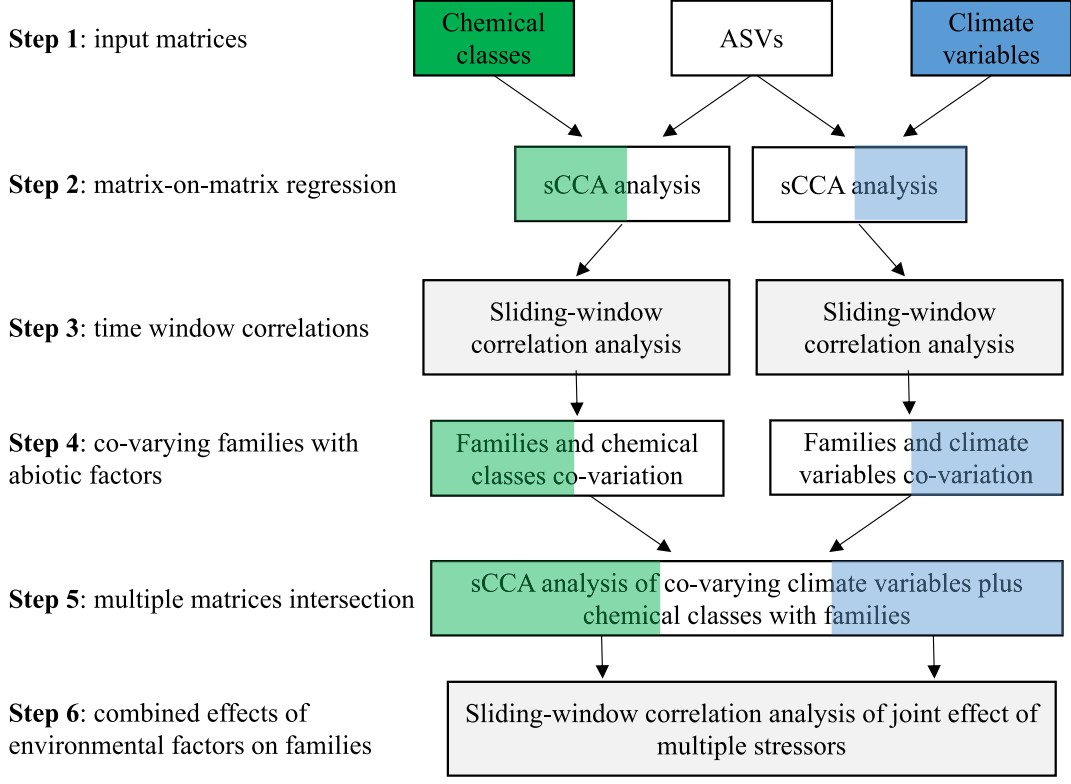

**Appendix 1—figure 5.** AI pipeline. The analytical pipeline consists of six main steps: Step 1 is the preparation of input data matrices (ASVs, biocides and climate variables) to be used in the sCCA analysis. The type of environmental data may vary with the study; Step 2 is the matrix-on-matrix regression between the ASVs and another environmental data matrix, biocides or climate in this study. Following the sCCA analysis, the ASVs are assigned to family level (or other relevant taxonomic group); Step 3 consists of a Sliding Window (Pearson) Correlation (SWC) analysis, used to identify significant temporal correlations between families and environmental variables from the sCCA analysis; Step 4 identifies the families that co-vary with either biocides or climate variables independently; Step 5 is used to perform an intersection analysis among multiple matrices (families, biodices and climate variables); Step 6 applies a Sliding Window (Pearson) Correlation (SWC) analysis to identify families, whose relative abundance changes both with biocides and climate variables over time. The pipeline enables the ranking of environmental variables or their combination thereof that is inversely correlated to the relative abundance of families over time.

