## [Editor Report · eLife assessment]

This **fundamental** study advances the analytic toolset and understanding of long-term series of biological (freshwater) communities, and the impact of humans on these. The authors highlight the value of including not only spatiotemporal scales in biodiversity assessments but also some of the possible drivers of biodiversity loss. Analyzing their joint contribution as environmental stressors, the authors provide **compelling** evidence that ecosystem assessment methods currently used by environmental regulators throughout Europe are not fit-for-purpose, and they identify several alternatives, more robust indicators of freshwater ecosystem health. The work is timely and will be of interest to ecologists, modelers and global warming scientists in general.

---

## [Referee Report · Joint Public Review]

Introduction - Well written and placed within the current trends of unprecedented biodiversity loss, with emphasis in freshwater ecosystems. The authors identify three important points as to why biodiversity action plans have failed. Namely, community changes occur over large spatio-temporal scales and monitoring programs capture a fraction of these long-term dynamics (e.g. few decades) which although good at capturing trends in biodiversity change, they often fail at identifying the drivers of these changes. Additionally, most of these rely on manual sorting of samples, overlooking cryptic diversity, or state-of-the-art techniques such as sedimentary DNA (sedaDNA) which allow studying decade-long dynamics, usually focus on specific taxonomic groups unable to represent community-level changes. Secondly, the authors identify that biodiversity is threatened by multiple factors and are rarely studied in tandem. Finally, the authors stress the need for high-throughput approaches to study biodiversity changes since historically, most conservation efforts rely on highly specialized skills for biodiversity monitoring, and even well studied species have relatively short time series data. The authors identify a model freshwater lake (Lake Ring, Denmark) - suitable due to its well documented history over the last 100 years - to present a comprehensive framework using metabarcoding, chemical analysis and climatic records for identifying past and current impacts on this ecosystem arising from multiple abiotic environmental stressors.

Results - They are brief and should expand some more. Particularly, there are no results regarding metabarcoding data (number of reads, filtering etc.). These details are important to know the quality of the data which represents the bulk of the analyses. Even the supplementary material gives little information on the metabarcoding results (e.g. number of ASVs - whether every ASV of each family were pooled etc.). The drivers of biodiversity change section could be restructured and include main text tables showing the families positively or negatively correlated with the different variables (akin to table S2 but simplified).

Discussion

The discussion is well written identifying first, some of the possible caveats of this study, particularly regarding classification of metabarcoding data, its biases and the possible DNA degradation of ancient sediment DNA. The authors discuss how their results fit to general trends showing how agricultural runoff and temperature drive changes in freshwater functional biodiversity primarily due to their synergistic effects on bioavailability, adsorption, etc. The authors highlight the advantage of using a system-level approach rather than focusing on taxa-specific studies due to their indicator status. Similarly, the authors justify the importance of studying community composition as far back as possible since it reveals unexpected patterns of ecosystem resilience. Lake Ring, despite its partially recovered status, has not returned to its semi-pristine levels of biodiversity and community assemblage. Additionally, including enzyme activity allows to assess functional diversity of the studied environment although reference databases of these pathways are still lacking. Finally, the authors discuss the implications of their findings under a conservation and land management framework suggesting that by combining these different approaches, drivers of biodiversity stressors can be derived with high accuracy allowing for better informed mitigation and conservation efforts.

---

## [Author Response]

The following is the authors’ response to the original reviews.

We thank the reviewers for the constrictive and detailed feedback provided. We have adopted the proposed changes to improve the manuscript clarity and accessibility. The following revisions are included in the revised manuscript:

**Reviewer #1 (Public Review):**
The analytical framework is not sufficiently explained in the main text.

We think the reviewer is referring to the conceptual framework mentioned in introduction. In the previously submitted manuscript, we did not provide details because the framework is published elsewhere. However, we agree with the reviewer that a short explanation may be helpful, which we have included in the resubmitted manuscript.

The significance of findings in relation to functional changes is not clear. What are the consequences of enrichment of RNA transport or ribosome biogenesis pathways between pesticides and recovery stages, for example?

We thank the reviewer for this suggestion. In the previously submitted manuscript, we included an explanation of the central functions these pathways can alter (e.g. metabolism and infection response). These functions are self-explanatory. However, we have elaborated on the consequence that the disruption of these pathways can cause in the resubmitted manuscript.

The impact of individual biocides and climate variables, and their additive effects, are assessed but there is no information offered on non-additive interactions (e.g., synergistic, antagonistic).

This was a misunderstanding based on our use of the term synergistic in this context. The approach by which we define a synergistic or joint effect of two environmental variables on a taxonomic group is explained in the methods section. This analysis is based on climate variables and biocide types contributing the largest covariances in the correlation analysis explained in Supplementary Fig. 5; Step 4. The combined effect of two environmental variables on a taxon was considered to be significant if the biocide type and the climate variable were each significantly correlated with the taxon over the same time window, and their average Pearson correlation was > 0.5 with padj < 0.05 (SWC analysis with 10,000 permutations). The biocide type and the climate variable were interpreted to have a joint effect on a given taxon if the linear combination of the biocide type and the climate variable had a larger Pearson correlation coefficient than each of the correlations between the family and the biocide type and the family and the climate variable individually, in the same time interval with padj < 0.05 (with 10,000 permutations in the SWC analysis). We realise that the use of synergistic or additive was not correct in this context and have replaced the term synergistic with joint effect throughout the manuscript.

The level of confidence associated with results is not made explicit. The reader is given no information on the amount of variability involved in the observations, or the level of uncertainty associated with model estimates.

As we didn’t use traditional statistical approaches, confidence level estimation in the traditional sense is not possible. Instead, we used permutation tests and adjusted P-values to identify significant correlations in the data. These approaches are more robust than traditional statistics for integrating and discovering complex, group-wise patterns among high-dimensional datasets. While most forms of machine learning require large sample sizes, sCCA uses fewer observations to identify the most correlated components among data matrices and captures the multivariate variability of the most important features.

The major implications of the findings for regulatory ecological assessment are missed. Regulators may not be primarily interested in identifying past "ecosystem shifts". What they need are approaches which give greater confidence in monitoring outcomes by better reflecting the ecological impact of contemporary environmental change and ecosystem management. The real value of the work in this regard is that: (1) it shows that current approaches are inappropriate due to the relatively stable nature of the indicators used by regulators, despite large changes in pollutant inputs; (2) it presents some better alternatives, including both taxonomic and functional indicators; and (3) it provides a new reference (or baseline) for regulators by characterizing "semi-pristine" conditions.

We thank the reviewer for this suggestion, which we have included in the main text (L451461)

**Reviewer #2 (Public Review):**
Results - They are brief and should expand some more. Particularly, there are no results regarding metabarcoding data (number of reads, filtering etc.). These details are important to know the quality of the data which represents the bulk of the analyses. Even the supplementary material gives little information on the metabarcoding results (e.g. number of ASVs - whether every ASV of each family were pooled etc.).

We thank the reviewer for this suggestion. We have included a paragraph in results reporting read numbers and other statistics. The filtering criteria and handling of samples can be found in methods (L658-661; L670-675). As explained in methods the taxonomy was assigned using qiime feature-classifier classify-sklearn and used at family level where possible. When classification was not possible at family level because of incomplete/missing information in the online database or a poor match to reference database, the lowest classification possible was used.

The drivers of biodiversity change section could be restructured and include main text tables showing the families positively or negatively correlated with the different variables (akin to table S2 but simplified).

As there are over 180 unique families/taxonomic units correlated with at least one biocide or environmental variable, a simplified version of this table would be too large to include in the main text. Therefore, we prefer to keep this information in supplementary table 2 complete with correlation statistics.

We thank the reviewers for providing detailed feedback on the manuscript and respond to their suggestions as follows:

**Reviewer #1 (Recommendations For The Authors):**
Thank you for the opportunity to review your manuscript, which I found interesting and enjoyable to read. Here are some suggestions for improving it.Remove spaces before citations in text.Lines 51-53: "Community-level biodiversity reliably explained freshwater ecosystem shifts whereas traditional quality indices (e.g. Trophic Diatom Index) and physicochemical parameters proved to be poor metrics for these shifts." Seems to be the wrong way around / not clear???

Rephrased to clarify.

Line 54: Should be "...advocates the use of..." or "...demonstrates the advantages of..."

Done, thanks for the suggestion.

Line 62: Spell out numbers <10, i.e. "sixth mass extinction"

Done, thank you.

Lines 66-72: These sentences lack clarity. It's not clear that "experimental manipulation of biodiversity" hasn't involved investigation of "multi-trophic changes". By the third of these four sentences it is not clear what "they" is referring to. And in the fourth sentence, "these holistic studies" are not defined. Perhaps it would suffice to say that experiments have so far focused primarily on a single trophic level and largely neglected freshwater systems.

We have rephrased to improve clarity.

Line 81: Delete unnecessary bracket

Done, thank you.

Line 82: "a minority of freshwater ecosystems" sounds as if you're saying that few freshwater ecosystems are represented in BioTIME, which seems obvious and would also apply to terrestrial and marine systems. Do you mean that freshwater ecosystems re not well represented in the data?

We have clarified the sentence, thanks.

Line 106: Resolve issue with citation in text at the end of the sentence (repeated at line 109 and possibly other lines).

Done, thank you.

Line 116: By ">1999s" do you mean 1990s?

This was a typo. it was supposed to be >1999

Line 120: The reader would benefit greatly from a brief explanation of explainable network models and multimodal learning in the introduction. Why are these the right tools to use? How do they work in this context? Figure 1 helps to some extent but needs more commentary in the text.

We have included an explanation of the explainable network models and multimodal learning and how their use can be beneficial to the study of diverse data types.

Line 144: Here and throughout the text the language could be much more efficient and readable. "Alpha diversity" does not require a definite article. Furthermore, when referring to significance it is convention to state the p-value, test statistic and test used.

As there are different p-values for each barcode, we have included them in legend to Supplementary Fig. 1 to avoid crowding the main text. We prefer to leave the text unchanged for this reason.

Line 155: "The primary producer's composition" is grammatically awkward and less suitable than "the composition of primary producers". This kind of awkwardness occurs again at line 285 ("diatom's") and possibly in other parts of the manuscript.

Thanks, corrected.

Line 169: The statement that this family was "relatively more abundant" needs a little more explanation. What is it relative to - other groups or to previous stages?

More abundant than in the other phases – the sentence has been modified.

Line 179: Nested brackets are unnecessary and affect readability. This could simply be a new sentence, i.e. "For example, Nitrospiraceae (nitrite oxidizers)..."

Done, thanks.

Line 215: "Functional biodiversity", which implies that some biodiversity is functional and some not, does not seem an appropriate term to describe the results you present in this section. Simply "functioning of the prokaryotic community" would suffice.

Thanks, done.

Line 214-233: This section may be inaccessible for many readers. For example, what are Kegg Orthologs and what role do they play in the functioning of a lake ecosystem? The explanation comes later in the paragraph but there needs to be a gentler introduction before diving into specific technical concepts.

We appreciate this comment and have included a short explanation of what KEGG and KO terms mean.

Supplementary Figure 3: It would be helpful to superimpose the lake stages here, as done in Figure 2.

The figure has been updated with coloured data points corresponding to each phase, as in supplementary figure 1.

Line 265: Should be "19 of which were identified..."

Done, thanks.

Line 284: "Predominantly" rather than "prominently"?

Done

Line 242-316: This section is good in that it identifies and ranks individual biocides and climate variables but there is no information on non-additive interactions (e.g., synergistic, antagonistic). Could the authors at least comment on why this was not done or not necessary, and what uncertainties this omission could introduce into the results?

This was a misunderstanding based on our use of the term synergistic in this context. the approach by which we define a synergistic or joint effect of two environmental variables on a taxonomic group is explained in the methods section. This analysis is based on climate variables and biocide types contributing the largest covariances in the correlation analysis explained in Supplementary Fig. 5; Step 4. The combined effect of two environmental variables on a taxon was considered to be significant if the biocide type and the climate variable were each significantly correlated with the taxon over the same time window, and their average Pearson correlation was > 0.5 with padj < 0.05 (SWC analysis with 10,000 permutations) – this is shown in Supplementary Fig. 5; Step 6. The biocide type and the climate variable were interpreted to have an additive effect on a given taxon if the linear combination of the biocide type and the climate variable had a larger Pearson correlation coefficient than each of the correlations between the family and the biocide type and the family and the climate variable individually, in the same time interval with padj < 0.05 (with 10,000 permutations in the SWC analysis). we have replace synergistic with joint effect to avoid confusion.

Figure 4: These 3-D plots are very hard to read. Without additional features (e.g. shadows on each plane, or lines connecting points to planes) it is impossible for the viewer to tell where the points are located on each axis.

We have created interactive 3D plots here: https://environmental-omicsgroup.github.io/Biodiversity_Monitoring/.

Figure 5: Legend entry should be "summer precipitation" not "precipitations". "Additive effect" rather than "joint effect" would be more consistent with the main text.

“Precipitations” has been updated to “precipitation” where relevant throughout. We left ‘joint effect’ and unified the main text, responding to a previous comment of this reviewer on the meaning of synergistic effects in our study.

Line 348: Doesn't your approach also require specialist skills? I often feel that the "traditional" versus "molecular" monitoring debate misses this point. Some comment on the training and development needs for those interested in applying the sedaDNA approach would be welcome. Otherwise it is an unfair comparison.

Whereas the application of high throughput sequencing technologies requires training, these technologies are well established with publicly available standard operating procedures. As compared to direct observations, high throughput sequencing provides replicable results regardless of the operator. Moreover, the application of metabarcoding to sedaDNA or more generally eDNA can be outsourced to established environmental services, removing the need for training if it is a limiting factor. The above has been included in discussion.

Line 391: "Significantly did" what? "Did significantly change over time" would be better.

Done, thanks.

Line 407: Should be "an indicator of..." and "did not significantly change over time..."

Done, thanks.

Line 408-410: Regulators are not necessarily interested in identifying past "ecosystem shifts", so this does not seem to be the best way to contrast the capabilities of the sedaDNA approach with those of LTDI2. The real value of this work, in my opinion, is threefold. First, it shows that the reliance on diatoms as indicators of ecological status is inappropriate due to the relatively stable nature of diatom communities in the face of large environmental changes. Second, it presents some better alternatives, including both taxonomic and functional indicators. And third, it provides a new reference point for regulators by characterising "semi-pristine" conditions.

Thanks for the insightful suggestion. We agree with the reviewer on the advantages and have spelled them out in the resubmitted manuscript.

Line 445: What are "housekeeping functions"? I checked the Cuenca-Cambronero paper cited but did not find the term there.

Housekeeping functions are essential basic cellular functions that are evolutionary conserved. They are more commonly present in public databases because they have been characterised in a number of model species (e.g. Drosophila, *C. elegans* and Mus musculus). Our reference it not to the Cuenca-Cambronero paper, but to Mi et al, describing the reference database PANTHER. We included the definition of housekeeping functions in the main text.

Line 449: Briefly state the main functional changes found here.

Examples have been included.

Lines 451-452: Whilst this statement may be found in the cited source, most readers I suspect would not identify with it. Indeed, one could argue that most of freshwater ecology has been dedicated to this very task (documenting chemical impacts on biodiversity)! A more balanced view is needed here.

The sentence the reviewer refers to includes also reference to climate change. Climate change and chemical pollution are the two most common causes of biodiversity loss, and not only in freshwater ecosystems.

Lines 463-466: These examples both point to non-additive (synergistic) effects, which were not assessed in the current study.

Please refer to our explanation above about the inappropriate use of synergistic and, here, additive. We have altered the text throughout to use joint effects as we do not investigate synergistic, antagonistic and additive effects as traditionally described in ecology.

Lines 472-474: This sentence is unclear. Do you mean that this approach surpasses others in terms of reliability? If so, I don't believe this has been demonstrated in the paper.

We apologise. The word ‘reliability’ should have not been in the text. We have improved the clarity of this sentence.

Lines 474-482: In these sentences it is unclear whether or not you are talking about your method or contrasting it with another method(s). If the latter, which method or methods are you referring to?

We have fixed this sentence to better reflect that our algorithm provides a high degree of confidence that surpasses state-of-the-art analysis, which predominantly identify patterns of co-occurrence of taxa within communities (e.g. Correlation-Centric Network).

Line 631: Should be "Physico-chemical variables". I have not extensively checked the rest of the methods for such errors.

Thank you, the text has been changed where present.

**Reviewer #2 (Recommendations For The Authors):**
IntroductionLine 80 remove extra ''

Done, thank you.

Line 81 rephrase e.g includes few freshwater ecosystems

We modified this sentence also following Reviewer #1

Line 83 although, instead of whereas?

Done, thanks.

Line 106 formatting reference issueLine 109 same as above

Thank you, noted.

ResultsLine 141 - 144 how was the sampling of the sediment performed over the 100 year core? Every year? Every 5 years? Or were they pooled to represent the (as of yet unlisted) phases?

The reviewer is correct that details are not provided here. They are in methods. We have added some text to explain the basic concepts of how the core was obtained and sliced and refer the reader to the method section for more details.

Line 154 the authors have not yet explicitly listed the lake phases, so it is difficult to refer to them now.

Noted, the addition of a short explanation at the beginning of the results section should take care of this issue.

Line 216 - may be worth briefly explaining KEGG orthologs and how these relate to functional biodiversity.

We thank the reviewer. Also responding to a similar comment from Reviewer #1, we included a description of KO terms and their links to functional biodiversity.

Lines 249 - 260 instead of a supplementary table, it could remain in the main text

Supplementary table 2 is a multi-tab table including information for each region amplified here. It is not possible to include this table in the main text.

Materials and MethodsDue to the formatting of the manuscript (results & discussion before materials and methods), many of the results are not clearly understood without having to visit the M&M section. Particularly, how the biocide types were obtained (Historic records plus persistence of DDT in sediments). This could be resolved y including a few sentences on how the data was gathered in the results section. Overall, materials and methods are sufficient, however, it is not clear how many of the 37 metabarcoding samples correspond to which of the lake phases. Finally, I suggest a better organization of M&Ms by having subheadings for each section. For example, under Biodiversity fingerprinting across 100 years, one subheading could de DNA extraction and sequencing, another subheading could be bioinformatics.

We thank the reviewer for the suggestion. To alleviate the issues linked to the methods section coming after the results section, we have introduced a short explanation of the sediments core and the lake phases at the beginning of the results section. A description of the climate and chemical data has been included at the beginning of the section ‘Drivers of biodiversity change’ in results. Subheadings were introduced in methods as suggested.